# Control of endothelial cell polarity and sprouting angiogenesis by non-centrosomal microtubules

**Maud Martin[1], Alexandra Veloso[2,3], Jingchao Wu[1], Eugene A Katrukha[1], Anna Akhmanova[1]\***

[1]Cell Biology, Department of Biology, Faculty of Science, Utrecht University, Utrecht, Netherlands; [2]Interdisciplinary Cluster for Applied Genoproteomics, University of Liège, Liège, Belgium; [3]GIGA-Molecular Biology in Diseases, University of Liège, Liège, Belgium

**Abstract** Microtubules control different aspects of cell polarization. In cells with a radial microtubule system, a pivotal role in setting up asymmetry is attributed to the relative positioning of the centrosome and the nucleus. Here, we show that centrosome loss had no effect on the ability of endothelial cells to polarize and move in 2D and 3D environments. In contrast, non-centrosomal microtubules stabilized by the microtubule minus-end-binding protein CAMSAP2 were required for directional migration on 2D substrates and for the establishment of polarized cell morphology in soft 3D matrices. CAMSAP2 was also important for persistent endothelial cell sprouting during in vivo zebrafish vessel development. In the absence of CAMSAP2, cell polarization in 3D could be partly rescued by centrosome depletion, indicating that in these conditions the centrosome inhibited cell polarity. We propose that CAMSAP2-protected non-centrosomal microtubules are needed for establishing cell asymmetry by enabling microtubule enrichment in a single-cell protrusion.

DOI: https://doi.org/10.7554/eLife.33864.001

**\*For correspondence:**
a.akhmanova@uu.nl

## Introduction

Cell polarization is a prerequisite for virtually every specialized cellular process. By analogy with the compass on a ship navigating to its destination, the centrosome has been assumed to play a central role in governing cell polarity (*Bornens, 2012*), including two-dimensional (2D) mesenchymal migration, in which cells are organized into an extending leading edge and a contractile cell rear (*Ridley et al., 2003*; *Tang and Marshall, 2012*). The widely accepted dogma connecting the centrosome and the direction of cell movement originated from the observation that during migration, the centrosome is typically positioned at the front of the nucleus (*Gotlieb et al., 1981*; *Koonce et al., 1984*; *Kupfer et al., 1982*; *Malech et al., 1977*). Such centrosome positioning is observed in different systems (*Elric and Etienne-Manneville, 2014*; *Luxton and Gundersen, 2011*), although it does not apply to certain specific cell types (*Luxton and Gundersen, 2011*; *Yvon et al., 2002*) or in the presence of environmental constraints (*Doyle et al., 2009*; *Pouthas et al., 2008*; *Schütze et al., 1991*). Despite the scarcity of direct experimental evidence, currently based on laser ablation of the pericentrosomal area (*Koonce et al., 1984*; *Wakida et al., 2010*) or indirect interference with centrosome localization (*Dujardin et al., 2003*; *Etienne-Manneville and Hall, 2001*; *Levy and Holzbaur, 2008*; *Schmoranzer et al., 2009*), the orientation of the nucleo-centrosomal axis is commonly regarded as a major cell polarity determinant.

Microtubules (MTs) are thought to support cell polarity by forming an asymmetrical network (*Etienne-Manneville, 2013*). This asymmetry can potentially be generated by specific positioning of the

**eLife digest** Networks of blood vessels grow like trees. Sprouts appear on existing vessels, stretching out to form new branches in a process called angiogenesis. The cells responsible are the same cells that line the finished vessels. These "endothelial cells" start the process by reorganizing themselves to face the direction of the new sprout, changing shape to become asymmetrical, and then they begin to migrate.

Beneath the surface, a network of protein scaffolding supports each migrating cell. The scaffolding includes tube-like fibers called microtubules that extend towards the cell membrane and organize the inside of the cell. Destroying microtubules damages blood vessel formation, but their exact role remains unclear.

A structure called the centrosome can organize microtubules within cells. The centrosome was generally believed to act like a compass, pointing in the direction that the cell will move. Microtubules can anchor to the centrosome, and this structure is thought to play an important role in cell migration. Yet, many microtubules organize without it; these microtubules instead are organized by a compartment of the cell called the Golgi apparatus and are stabilized by a protein named CAMSAP2.

Martin et al. now report that removing the cells' centrosomes did not affect cell migration, but getting rid of CAMSAP2 did. Analysis of cell shape and movement in cells grown in the laboratory and in living animals revealed that cells cannot become asymmetrical, or "polarize", and migrate without CAMSAP2.

In a two-dimensional wound-healing assay, a sheet of cells originally grown from the vessels of a human umbilical cord was scratched, and a microscope was then used to record the cell's movement as they repaired the injury. Normally, the cells on either side move in a straight line using their microtubules, and though the process was not affected in cells without centrosomes, it was in those without CAMSAP2.

Even more striking results were seen in three-dimensional assays. When the same blood vessel cells from human umbilical cords are grown as spheres inside collagen gels, they form sprouts as they would in the body. Without CAMSAP2, the cells could not organize their microtubules and they were unable to elongate in one direction and form stable sprouts. Lastly, depleting CAMSAP2 also prevented the normal formation of blood vessels in zebrafish embryos.

Taken together, these findings change our understanding of how microtubules affect cell movement and how important the centrosome is for this process. Further work could have an impact on human health, not least in cancer research. Tumors need a good blood supply to grow, so understanding how to block blood vessel formation could lead to new treatments. Microtubules are already a target for cancer therapy, so future work could help to optimize the use of existing drugs.

DOI: https://doi.org/10.7554/eLife.33864.002

centrosomal anchor of a radial MT network relative to other cell structures, such as the nucleus, but also by MTs that do not originate from the centrosome (*Akhmanova and Hoogenraad, 2015*; *Alieva et al., 2015*; *Vinogradova et al., 2009*). The minus ends of centrosome-independent MTs can associate with the members of the calmodulin-regulated spectrin-associated protein (CAMSAP)/Patronin/Nezha family (*Akhmanova and Hoogenraad, 2015*). In mammalian cells, CAMSAP2, together with CAMSAP3 in certain cell types, binds to free, non-centrosomal MT minus ends and promotes their stabilization (*Jiang et al., 2014*; *Tanaka et al., 2012*). Interestingly, CAMSAP2 has recently been shown to participate in MT stabilization at the Golgi apparatus (*Wu et al., 2016*), a site that can function as an alternative MT-organizing centre (MTOC) (*Zhu and Kaverina, 2013*). In mammalian cells, the Golgi is often positioned close to the centrosome (*Rios, 2014*), and thus locates in front of the nucleus during cell migration (*Koonce et al., 1984*; *Kupfer et al., 1982*; *Malech et al., 1977*). As the central organizer of the secretory pathway, an anteriorly positioned Golgi is thought to support polarized transport needed to sustain directional migration (*Yadav and Linstedt, 2011*). Consistently, Golgi disorganization without MT disassembly (*Bisel et al., 2008*; *Hurtado et al., 2011*; *Yadav et al., 2009*), as well as the loss of Golgi-associated MTs (*Hurtado et al., 2011*; *Miller et al., 2009*) prevent proper polarized Golgi trafficking without

affecting global secretory properties but lead to defects in directional cell movement. Notably, investigation of the contributions of centrosomal and Golgi-originating MT populations to Golgi organization indicated that the role of the centrosome was restricted to facilitating assembly of an integral Golgi apparatus (*Vinogradova et al., 2012*).

During angiogenesis, the process of new blood vessel development, endothelial cells (ECs) respond to external cues by coordinating numerous activities, including proliferation, sprouting, migration, lumen formation and anastomosis (*Geudens and Gerhardt, 2011*; *Potente et al., 2011*). The first step of vessel formation, outward cell sprouting, requires that the tip cell, which will guide growth of a new vessel, breaks symmetry by extending protrusions toward guidance cues (*Geudens and Gerhardt, 2011*; *Lee and Bautch, 2011*). MT growth dynamics has been shown to be required for the formation and maintenance of angiogenic structures (*Bayless and Johnson, 2011*; *Myers et al., 2011*), whereas its regional regulation has been implicated in directional EC migration (*Braun et al., 2014*). Vascular sprouting and repolarization are also affected by supernumerary centrosomes, a hallmark of tumor ECs that impacts on MT nucleation and dynamics (*Kushner et al., 2014*; *Kushner et al., 2016*).

The ability of ECs to polarize can be explored in 2D, where cells respond to a monolayer wound by developing a mesenchymal motile phenotype typified by a front-rear asymmetry and anterior centrosome positioning (*Gotlieb et al., 1981*). When cultured in 3D collagen gels, ECs form branched tubular structures (*Koh et al., 2008*), and when spheroids of ECs are embedded in a 3D matrix, they develop sprouts that closely reproduce the first step of formation of capillaries from pre-existing vessels (*Pfisterer and Korff, 2016*).

Here, we used a combination of 2D and 3D endothelial models to investigate the role of MT organization in cell polarization. Challenging the prevailing view, our data showed that the centrosomal MT population is dispensable and insufficient for EC migration and sprouting. In contrast, silencing of CAMSAP2, which results in disappearance of non-centrosomal MTs, profoundly perturbed the ability of ECs to form vascular sprouts in vitro as well as in vivo, in zebrafish embryos. Detailed analysis showed that non-centrosomal MTs are required to allow MT redistribution in a single-cell protrusion and thus enable polarized trafficking, directional stabilization of protrusions and persistent migration.

## Results

### The centrosome is not essential for endothelial migration and sprouting

Using 2D monolayer wound healing assay, we confirmed the anterior position of the centrosome and the Golgi apparatus in migrating ECs and also extended this observation to tip ECs sprouting from a spheroid in 3D (*Figure 1—figure supplement 1A*). To assess the impact of centrosome removal on these processes, we took advantage of the Plk4 inhibitor centrinone, which prevents centriole duplication and leads to centrosome depletion (*Wong et al., 2015*). Efficient centrosome elimination was confirmed by staining with different markers of centrioles or pericentriolar material (PCM) (*Figure 1—figure supplement 1B*). Loss of centrosomes as focal points of MT organization was also visible by super-resolution imaging of the MT networks in fixed cells and by tracing growing EB3-GFP-positive MTs in live cells (*Figure 1A,B*). Importantly, the centrosomal arrangement of MTs in control cells was more obvious in EB3-GFP tracings than in MT images (*Figure 1A,B*), suggesting that the centrosome plays a more important role in nucleating new MTs than in anchoring their minus ends, and that even in control cells, a significant MT population is non-centrosomal. The perinuclear MT density was partly associated with the Golgi apparatus, which was mildly enlarged and less compact in centrinone-treated cells (*Figure 1—figure supplement 1C*). Nocodazole washout assays showed that the centrosome was the major nucleation site during MT reassembly in control cells, while this function was taken over by the Golgi membranes in centrinone-treated cells (*Figure 1—figure supplement 1D*). In spite of these differences, MT density, the density of growing, EB3-positive MT plus ends, parameters of MT plus end growth and the levels of different tubulin post-translational modifications were not affected (*Figure 1A,B,C*, *Figure 1—figure supplement 2A,B*), and in contrast to a centrosome excess (*Godinho et al., 2014*), the distribution and intensity of cell-cell adhesion markers was unchanged (*Figure 1—figure supplement 2D* ) (). The abundance of MT minus-end stabilizing protein CAMSAP2 was mildly but not significantly increased, and the

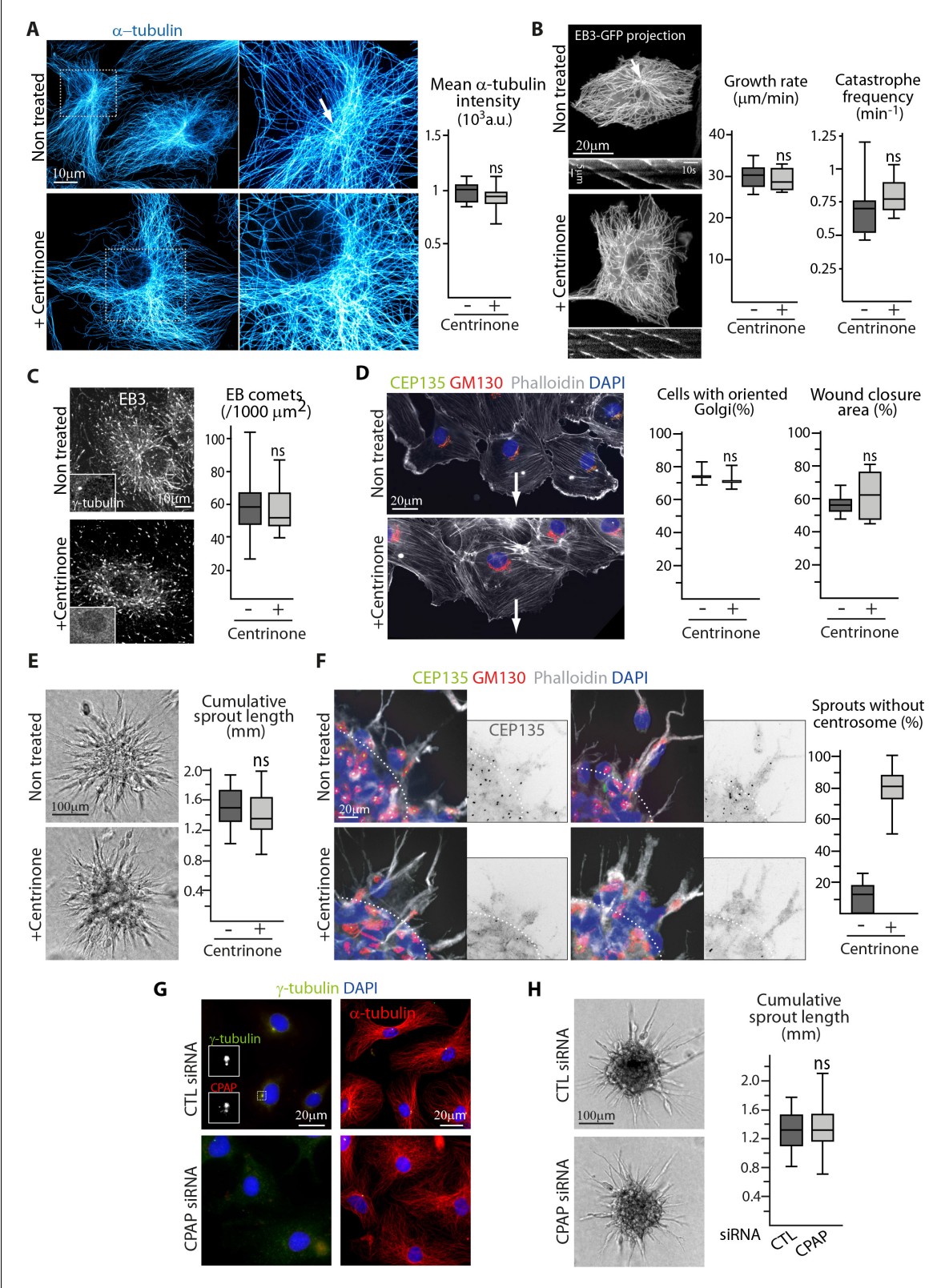

**Figure 1.** The centrosome is not essential for angiogenic migration and sprouting. (**A**) Imaging of control or centrinone-treated HUVECs stained for MT (α-tubulin, cyan hot) using STED microscopy. Arrow points toward the centrosome and the plot shows the average fluorescence intensity of α-tubulin, n = 25 cells for each condition. (**B,C**) Analysis of MT plus ends in control or centrinone-treated HUVECs illustrated by maximum intensity projections and kymographs of EB3-GFP live fluorescence imaging (**B**) and EB3 staining (**C**). Plots show MT growth rate and catastrophe frequency, n = 96 tracks in

*Figure 1 continued on next page*

*Figure 1 continued*

8 cells per condition (**B**), and the density of EB3 comets, n = 19 and 20 cells (**C**). Arrow points toward the centrosome. (**D,F**) Controlorcentrinone-treated HUVECs during a 2D monolayer wound healing assay (**D**) or during sprouting from a 3D spheroid (**F**) in the presence of thymidine, stained for the centriolar marker CEP135 (green), Golgi marker GM130 (red), F-actin (phalloidin, white) and DNA (DAPI, blue). Arrows point toward the wound in (**D**), and the dashed line indicates the position of the spheroid body in (**F**). Z-maximum projections of confocal images are shown on the left; the plot shows the proportion of cells with oriented Golgi, defined as being contained in the 90° sector facing the wound, n = 3 experiments including 158 and 132 cells in total, and quantification of the percentage of wound closure after 8 hr, n = 8 fields in two independent experiments for each condition (**D**) or the proportion of cellular sprouts devoid of centrosome, n = 10 spheroids representing 60 and 47 sprouts (**F**). (**E,H**) Spheroid sprouting assay with control, centrinone- (**E**) or CPAP siRNA-treated HUVECs (**H**) in the presence of thymidine. Representative micrographs are shown on the left; the plot shows quantification of the cumulative length of all sprouts per spheroid, n = 59 and 61 spheroids in four independent experiments (**E**) and n = 62 spheroids per condition in three independent experiments (**H**). (**G**) Staining of HUVECs transfected with control or CPAP siRNA for γ-tubulin (green), CPAP (red, left), MT (α-tubulin, red, right) and DNA (DAPI, blue). Data are shown using box plots; ns, no significant difference using Mann-Whitney U test (**A,B,C**, right plot in **D**) **E** or Student's unpaired two-tailed *t*-test (left plot in **D**).

DOI: https://doi.org/10.7554/eLife.33864.003

The following source data and figure supplements are available for figure 1:

**Source data 1.** An Excel sheet with numerical data on the quantification of the effect of centrinone treatment on the EC mean intensity of α-tubulin signal, MT dynamics parameters, EB comet number, the polarization of Golgi during migration, the efficiency of wound closure, the cumulative length of spheroid sprouts and the proportion of sprouting ECs with centrosome as well as the effect of CPAP depletion on the cumulative length of spheroid sprouts represented as plots in *Figure 1A–F,H*.

DOI: https://doi.org/10.7554/eLife.33864.006

**Figure supplement 1.** The centrosome is not essential for angiogenic migration and sprouting.

DOI: https://doi.org/10.7554/eLife.33864.004

**Figure supplement 1—source data 1.** An Excel sheet with numerical data on the quantification of the polarization of Golgi and centrosome during 2D migration and 3D sprouting as well as of centrosome removal efficiency using different (peri)centriolar markers, Golgi area and dispersion and MT nucleation activity and EB3 Golgi enrichment after nocodazole washout after centrinone treatment represented as plots (or as mean value ± SD for 1B) in *Figure 1—figure supplement 1A–D*.

DOI: https://doi.org/10.7554/eLife.33864.007

**Figure supplement 2.** The centrosome is not essential for angiogenic migration and sprouting.

DOI: https://doi.org/10.7554/eLife.33864.005

**Figure supplement 2—source data 1.** An Excel sheet with numerical data on the quantification of the effect of centrinone treatment on the expression of CAMSAP2 and various post-translationally modified tubulin in ECs, the mean intensity of acetylated tubulin signal, the density of CAMSAP2 stretches, the intensity of VE-Cadherin and ZO-1 signal at cell junctions, the velocity and directionality of cell migration during scratch-wound assays, as well as the effect of CPAP depletion on centrosome removal efficiency and the proportion of 3D sprouting ECs with centrosome represented as plots in *Figure 1—figure supplement 2A–G*.

DOI: https://doi.org/10.7554/eLife.33864.008

area occupied by CAMSAP2 stretches was enlarged (*Figure 1—figure supplement 2A,C*). This is consistent with the enlarged Golgi, to which many CAMSAP2 stretches attach (*Wu et al., 2016*), and possibly with an increased stabilization of non-centrosomal MTs that compensate for the loss of the centrosomal ones.

2D migration was not perturbed in centrinone-treated ECs (*Figure 1D*, *Figure 1—figure supplement 2E*). ECs in the migrating front still showed proper polarized organization with their Golgi positioned toward the direction of migration (*Figure 1D*). In a more physiological 3D context, centrosome removal did not affect the emergence of EC sprouts out of spheroids grown in collagen matrix (*Figure 1E*). The few remaining centrosome-containing ECs were not enriched in the sprouts, and the Golgi apparatus was properly polarized in sprouting tip ECs devoid of centrosomes (*Figure 1F*). To confirm these results, we depleted centrosomes by knocking down CPAP, a factor essential for centriole duplication (*Kohlmaier et al., 2009*; *Schmidt et al., 2009*; *Tang et al., 2009*). Also using this approach, we could efficiently remove centrosomes in the majority of cells without affecting EC polarization in 3D and sprouting from spheroids (*Figure 1G,H*, *Figure 1—figure supplement 2F,G*). These results indicate that centrosome is dispensable for endothelial polarization and movement.

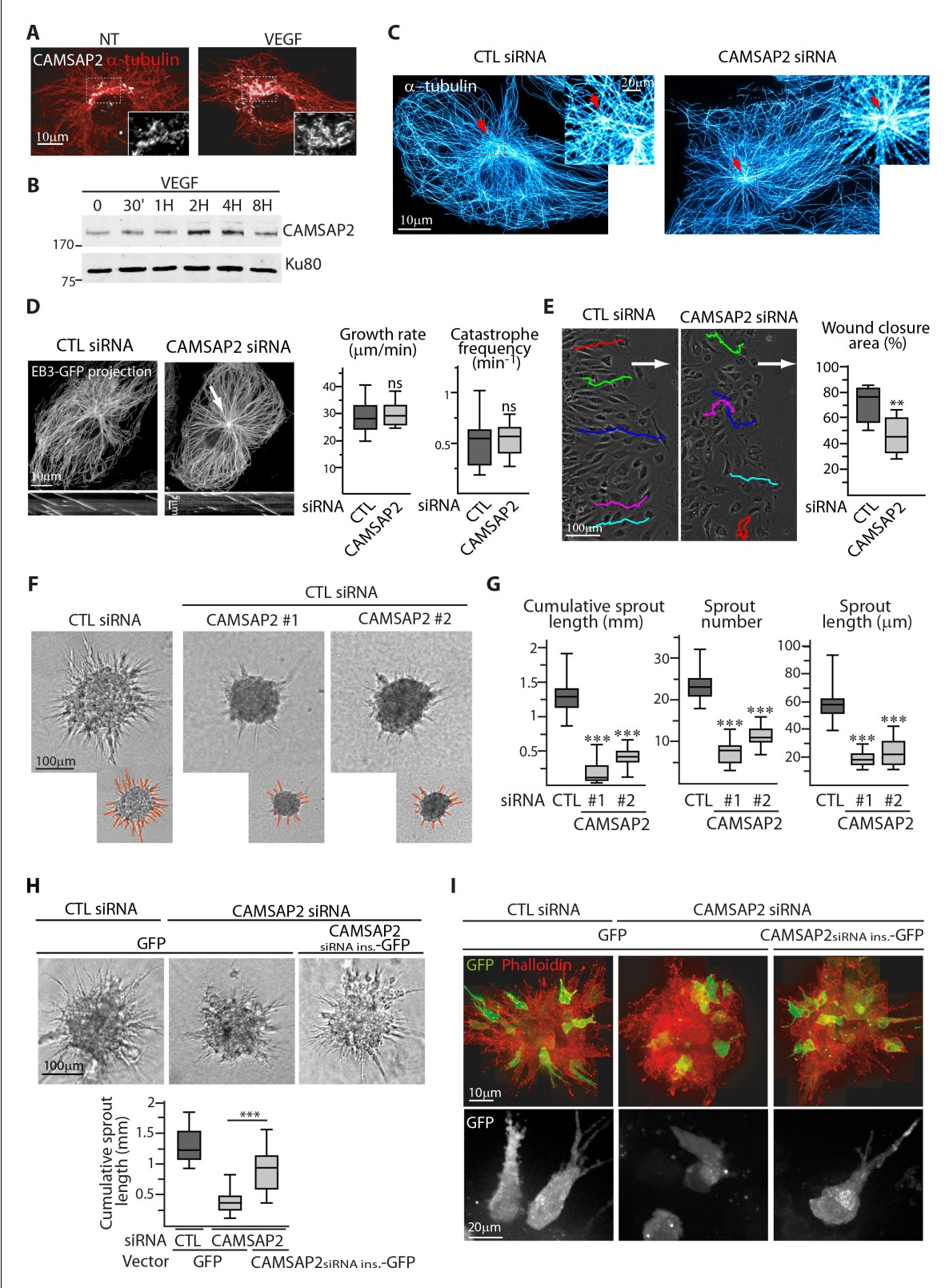

**Figure 2.** CAMSAP2 is required for maintaining non-centrosomal MTs and EC migration. (**A**) Staining of CAMSAP2 (white) and α-tubulin (red) in serum-starved HUVECs before or after a 2 hr treatment with VEGF. Wide-field fluorescence images are shown. (**B**) Western blots of total extracts of HUVECs during a VEGF stimulation experiment using antibodies against CAMSAP2 and Ku80 as loading control. (**C**) Imaging of control or CAMSAP2 siRNA-transfected HUVECs stained for MTs (α-tubulin, cyan hot) using STED microscopy. Arrows point toward the centrosome. (**D**) Analysis of MT dynamics in

*Figure 2 continued on next page*

*Figure 2 continued*

control or CAMSAP2 siRNA transfected HUVECs illustrated by maximum intensity projections and kymographs of EB3-GFP live fluorescence images. Plots show MT growth rate and catastrophe frequency, n = 132 tracks in 11 cells for each condition. Arrow points toward the centrosome. (E) Overlay of end-point phase-contrast images with the result of cell tracking after 12 hr of migration in a wound healing assay in control and CAMSAP2-depleted HUVECs. The arrows point the direction of migration into the wound. Plots show quantification of wound closure area after 8 hr of migration, n = 12 fields in three independent experiments. (F,G) Spheroids of HUVECs transfected with control or two independent siRNAs against CAMSAP2 were subjected to a sprouting assay for 24 hr. Bright-field micrographs (F) were used to quantify cumulative length of all sprouts per spheroid, their number and length (G); n = 65, 57 and 38 spheroids in three independent experiments. (H,I) HUVECs were co-transfected with control or CAMSAP2 siRNAs together with a GFP-tagged siRNA-insensitive mutant of CAMSAP2 or GFP alone. Quantification of sprouting was performed using bright-field micrographs (H), n = 39, 32 and 33 spheroids in two independent experiments. Live GFP imaging using confocal microscopy (bottom, Z-maximum projection) and phalloidin staining (top, Z-maximum projection) were performed 12 hr after spheroid sprouting (I). Data are shown using box plots; Mann-Whitney U test: ***p<0.001, **p<0.01, *p<0.05, ns, no significant difference.

DOI: https://doi.org/10.7554/eLife.33864.009

The following source data and figure supplements are available for figure 2:

**Source data 1.** An Excel sheet with numerical data on the quantification of the effect of CAMSAP2 silencing on MT dynamics parameters, the efficiency of wound closure, the cumulative length of spheroid sprouts, their number and average length, and the cumulative length of spheroid sprouts re-expressing CAMSAP2 represented as plots in *Figure 2D,E,G,H*.
DOI: https://doi.org/10.7554/eLife.33864.012

**Figure supplement 1.** CAMSAP2 is required for maintaining non-centrosomal MTs and cell migration in ECs.
DOI: https://doi.org/10.7554/eLife.33864.010

**Figure supplement 1—source data 1.** An Excel sheet with numerical data on the quantification of CAMSAP2 stretch number and length after VEGF treatment, as well as the effect of CAMSAP2 depletion on the EC mean intensity of α-tubulin signal, EB comet number, the expression of CAMSAP2 and various post-translationally modified tubulin, MT nucleation activity and EB3 Golgi and centrosome enrichment after nocodazole washout, the number, speed and length of KIF13B tracks and the velocity of cell migration during scratch-wound assays represented as plots in *Figure 2—figure supplement 1A,C,E,F,H,I*.
DOI: https://doi.org/10.7554/eLife.33864.013

**Figure supplement 2.** CAMSAP2 is required for maintaining non-centrosomal MTs and cell migration in ECs.
DOI: https://doi.org/10.7554/eLife.33864.011

**Figure supplement 2—source data 1.** An Excel sheet with numerical data on the quantification of the EC mitotic index and doubling time after CAMSAP2 depletion, the cumulative length of spheroid sprouts after CAMSAP2 depletion and treatment with thymidine and after CAMSAP2 and CAMSAP3 depletion represented as plot in *Figure 2—figure supplement 2A,B,D*.
DOI: https://doi.org/10.7554/eLife.33864.014

## CAMSAP2 is required for maintaining non-centrosomal MTs and cell migration in ECs

If the centrosomal MTs are dispensable for cell motility in ECs, their function must be taken over by non-centrosomal MTs, which in mesenchymal cells are stabilized by CAMSAP2 (*Jiang et al., 2014*). Interestingly, CAMSAP2 expression was transiently enhanced upon treatment with the angiogenic factor VEGF, and the length and number of CAMSAP2-decorated MT stretches was increased (*Figure 2A,B*, *Figure 2—figure supplement 1A*), suggesting that non-centrosomal MTs might play a role in angiogenesis. To test this idea, we silenced CAMSAP2 (*Figure 2—figure supplement 1B*), thereby generating a mostly centrosome-anchored, radial MT array (*Figure 2C,D*, *Figure 2—figure supplement 1D*). This treatment had no effect on the MT density or parameters of MT plus end dynamics (*Figure 2D*, *Figure 2—figure supplement 1C*). The abundance of post-translationally modified forms of tubulin was also unchanged (*Figure 2—figure supplement 1E*), showing that ECs are in this respect different from U2OS cells, where loss of detyrosinated MTs was seen upon CAMSAP2 depletion (*Jiang et al., 2014*). MT nucleation from the centrosome was unaffected, in line with fact that CAMSAP2 shows no colocalization with the centrosome (*Figure 2—figure supplement 1F, G*). Since MT properties were largely unaltered, the motility of the motor protein kinesin-3 KIF13B, which is known to play an important role in transporting VEGF receptor in ECs (*Yamada et al., 2014*), was also unchanged (*Figure 2—figure supplement 1H*).

CAMSAP2 depletion caused a moderate but significant decrease in cell migration in 2D (*Figure 2E*, *Figure 2—figure supplement 1I*). Strikingly, the defect was much more pronounced during 3D sprouting: ECs depleted of CAMSAP2 were able to form only shorts sprouts and form long protruding structures (*Figure 2F,G*). This strong impairment was not related to cell viability or cell cycle progression defects (*Figure 2—figure supplement 2A*), and CAMSAP2 depletion in non-

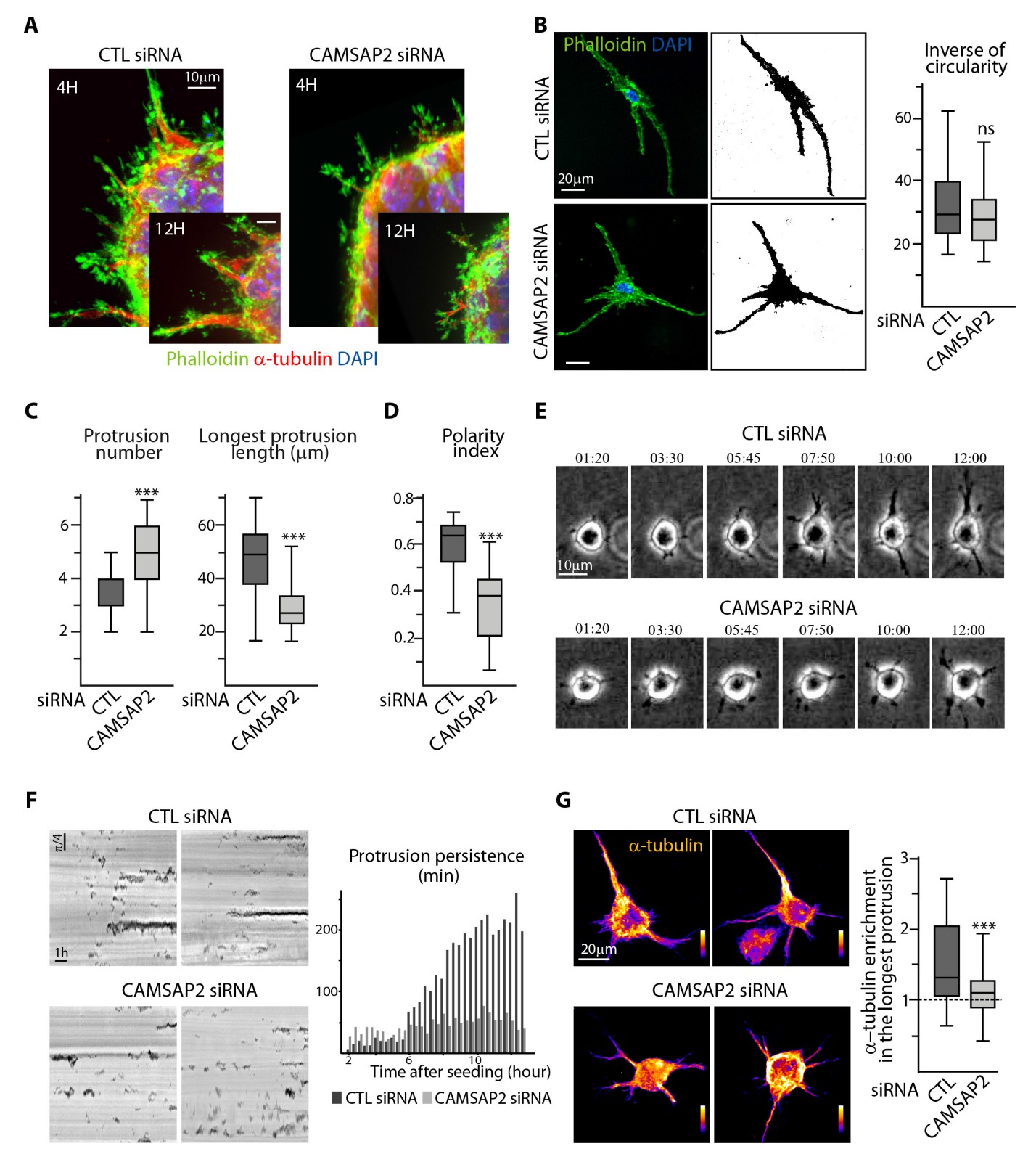

**Figure 3.** CAMSAP2 is required for stabilization of one major cell protrusion. (**A**) Staining for F-actin (phalloidin, green), α-tubulin (red) and DNA (DAPI, blue) in control or CAMSAP2-depleted sprouting spheroids. Z-maximum projections of confocal images are shown. (**B,C**) Staining for F-actin (phalloidin, green) and DNA (DAPI, blue) in 3D cultured control or CAMSAP2-depleted HUVECs. Z-maximum projections of confocal fluorescence images (left) were used to create binary cell masks using ImageJ (right) that were analyzed with ImageJ; plots show the inverse of circularity (representing the ratio

*Figure 3 continued on next page*

Figure 3 continued

between cell area and perimeter) (B), the total number of protrusions per cell and the length of the longest protrusion (C) in both conditions, n = 23 and 21 cells in three independent experiments. (D) Spatial distribution of protrusions in control and CAMSAP2-silenced 3D-cultured HUVECs. Polarity index was calculated as described in Materials ans methods to measure protrusion alignment with the major cellular axis: the index is close to one when the protrusions are polarized and align with the axis of the longest protrusion (small angle) and close to 0 when protrusions are dispersed (angle close to 90°), n = 23 and 21 cells in three independent experiments. (E) Live phase-contrast images of 3D protrusion dynamics in control and CAMSAP2-depleted HUVECs. Time is indicated in hr:min. (F) Radial resliced representation of the time-lapse described in (E). Signals indicate positions of protrusions at specific time points at specific radial positions. Plot shows quantification of the persistence of 3D protrusions formed by control or CAMSAP2-depleted HUVECs per 20 min interval. Data are mean from 12 cells in both conditions. (G) Staining for α-tubulin in 3D cultured control or CAMSAP2-depleted HUVECs. Z-maximum projections of confocal fluorescence images are shown using a color look-up table (LUT) and were used to calculate the average α-tubulin intensity ratio between the longest protrusion and the other ones, n = 18 cells for each condition. Data are shown using box plots; Mann-Whitney U test: ***p<0.001, ns, no significant difference.

DOI: https://doi.org/10.7554/eLife.33864.015

The following source data and figure supplements are available for figure 3:

**Source data 1.** An Excel sheet with numerical data on the quantification of the effect of CAMSAP2 depletion on the 3D elongation of ECs, the number of their 3D protrusions and the length of the longest one, their polarity index (protrusion organization), the persistence of the protrusions over time and the enrichment of α-tubulin signal in the longest protrusion represented as plots in *Figure 3B,C,D,F,G*.

DOI: https://doi.org/10.7554/eLife.33864.017

**Figure supplement 1.** CAMSAP2 is required for stabilization of one major cell protrusion.

DOI: https://doi.org/10.7554/eLife.33864.016

**Figure supplement 1—source data 1.** An Excel sheet with numerical data on the quantification of the effect of CAMSAP2 depletion on the number and length of spheroid protrusions over time, the total and average length of the 3D protrusions of isolated ECs, the binning of the average protrusion length by their direction, the polarity index of the 3D protrusions in relation to their length and the number of 3D protrusions over time represented as plots in *Figure 3—figure supplement 1A,C,E–G*.

DOI: https://doi.org/10.7554/eLife.33864.018

cycling ECs still severely affected formation of long sprouts (*Figure 2—figure supplement 2B*). Rescue experiments using expression of a siRNA-insensitive CAMSAP2 construct confirmed the specificity of the phenotype (*Figure 2H*), with ECs positive for the rescue construct populating growing sprouts in a CAMSAP2 knockdown background (*Figure 2I*). We also analyzed the potential involvement of CAMSAP3 and found that it was only weakly expressed in ECs and its depletion did not aggravate the phenotype of CAMSAP2 knockdown (*Figure 2—figure supplement 2C–E*). These data point to an important and specific role of CAMSAP2 in EC morphology in 3D.

## CAMSAP2 is required for stabilization of one major cell protrusion

To understand the poor ability of CAMSAP2-depleted cells to form long sprouts in 3D, we set out to characterize this process in more detail. We found that early stages of sprout formation were not affected much by CAMSAP2 depletion; however, the differences gradually increased over time, as, in contrast to the control situation, spheroids silenced for CAMSAP2 were unable to increase the number and especially the length of the sprouts (*Figure 3—figure supplement 1A*). In the absence of CAMSAP2, endothelial spheroids are thus capable of initiating protrusions but are unable to mature them into larger and more stable, MT-populated structures (*Figure 3A*).

In line with this idea, when individual isolated ECs were cultured in a collagen matrix, where they extended protrusions in different directions and fused into a tubulogenic network, CAMSAP2 depletion did not prevent the establishment of a vascular plexus (*Figure 3—figure supplement 1B*). Protrusive activity of isolated ECs measured by their elongation (inverse of circularity) and the total and average protrusion length was not affected by CAMSAP2 knockdown (*Figure 3B*, *Figure 3—figure supplement 1C*). However, such CAMSAP2-depleted ECs bearing a centrosome-centered MT array (*Figure 3—figure supplement 1D*) had a different organization of protrusions. Whereas control ECs had a restricted number of protrusions, with a single predominant one, CAMSAP2-depleted ECs displayed multiple short protrusions (*Figure 3B,C*). In contrast to control ECs that had most of their protrusions, and especially the longest ones, aligned in one direction, CAMSAP2-depleted cells displayed protrusions that were more radially dispersed, irrespective of their length (*Figure 3D*, *Figure 3—figure supplement 1E,F*).

To understand the origin of this phenotype, we performed live recording of protrusion formation. In contrast to control ECs, which, after having formed several small transient protrusions, stabilized

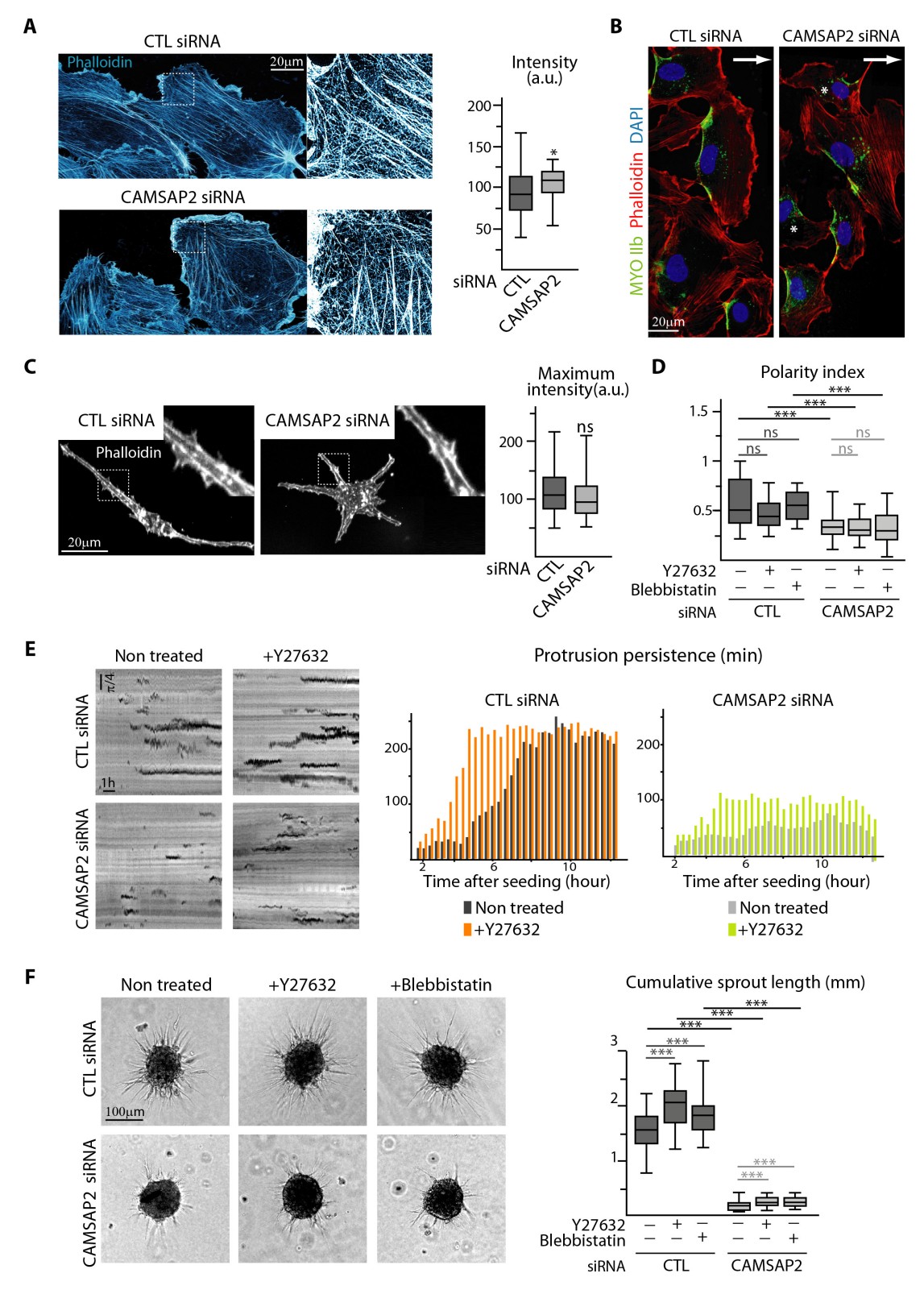

**Figure 4.** CAMSAP2 depletion phenotypes cannot be explained by changes in the actin cytoskeleton and cell contractility. (**A**) Imaging of control or CAMSAP2 siRNA-treated HUVECs during 2D wound healing assay stained for F-actin (phalloidin, cyan hot) using STED microscopy. The plot shows the average fluorescence intensity of phalloidin, n = 30 cells in two independent experiments for each condition. (**B**) HUVECs during 2D wound healing assay stained for Myosin IIb (MYOIIb, retracting edge marker, green), F-actin (phalloidin, red) and DNA (DAPI, blue). Z-maximum projections of confocal

*Figure 4 continued on next page*

*Figure 4 continued*

images are shown; the arrow points toward the wound and asterisks show disoriented HUVECs. (C) Staining for F-actin (phalloidin, white) in 3D cultured control or CAMSAP2-depleted HUVECs. Z-maximum projections of confocal fluorescence images are shown and fluorescence intensity profiles across protrusions were used to quantify the maximum intensity of phalloidin in protrusion (plot) as described in the Materials and methods, n = 40 cells in three independent experiments per condition. (D) Spatial distribution of protrusions in control and CAMSAP2-silenced 3D-cultured HUVECs treated or not treated with the contractility inhibitors Y-27632 or blebbistatin. Polarity index was calculated as described in Materials and methods to measure protrusion alignment with the major cellular axis, n = 26 cells in two independent experiments, except for blebbistatin treatment, where n = 20 cells. (E) Radial resliced representation of 3D protrusion dynamics in control and CAMSAP2-depleted HUVECs treated or not with Y-27632. Signals indicate positions of protrusions at specific time points at specific radial positions. Plot shows quantification of the persistence of the 3D protrusions. Data are mean from 12 cells in two independent experiments in both conditions. (F) Spheroids of HUVECs treated as in (D) were subjected to a sprouting assay for 24 hr. Bright-field micrographs are shown and were used to quantify cumulative length of all sprouts per spheroid; n = 43, 45, 47, 54, 51 and 54 spheroids in two independent experiments. Data are shown using box plots; Mann-Whitney U test: ***p<0.001, *p<0.05, ns, no significant difference.
DOI: https://doi.org/10.7554/eLife.33864.019

The following source data and figure supplements are available for figure 4:

**Source data 1.** An Excel sheet with numerical data on the quantification of the effect of CAMSAP2 depletion on the intensity of phalloidin signal in 2D (mean intensity) and in 3D (maximum intensity) ECs, as well as on the EC polarity index, the persistence of the protrusions over time and the cumulative length of spheroid sprouts after Y27632 and blebbistatin treatment represented as plots in *Figure 4A,C–F*.
DOI: https://doi.org/10.7554/eLife.33864.022

**Figure supplement 1.** CAMSAP2 depletion phenotypes cannot be explained by changes in the actin cytoskeleton and cell contractility.
DOI: https://doi.org/10.7554/eLife.33864.020

**Figure supplement 1—source data 1.** An Excel sheet with numerical data on the quantification of the effect of CAMSAP2 depletion on the proportion of coverage and the width of lamellipodia, the cumulative length and the width of stress fibers in 2D migrationg ECs, the activation level of Rho and Rac1 GTPases, the intensity of VE-Cadherin and ZO-1 signal at cell junctions and the intensity of phalloidin signal in 3D represented as plots (or mean value ± SD for 1B) in *Figure 4—figure supplement 1A–D*.
DOI: https://doi.org/10.7554/eLife.33864.023

**Figure supplement 2.** CAMSAP2 depletion phenotypes cannot be explained by changes in actin cytoskeleton and cell contractility.
DOI: https://doi.org/10.7554/eLife.33864.021

**Figure supplement 2—source data 1.** An Excel sheet with numerical data on the quantification of the effect of CAMSAP2 depletion and Y632 or blebbistatin treatment on the cumulative and average length of 3D protrusions from isolated EC, their number and the length of the longest ones as well as on the number and length of spheroid sprouts represented as plot in *Figure 4—figure supplement 2B–F*.
DOI: https://doi.org/10.7554/eLife.33864.024

and extended one or two of them, CAMSAP2-depleted ECs were unable to accomplish this transition and to elongate in a single direction (*Figure 3E*). Quantification of protrusion dynamics using a radial reslice representation (*Figure 3F*; every black signal represents the presence of a protrusion at a given time point at a given radial position) confirmed these observations and pointed to dramatically lower protrusion persistence after CAMSAP2 inactivation (*Figure 3F*), while the total protrusion number was not affected by CAMSAP2 depletion (*Figure 3—figure supplement 1G*).

We hypothesized that in CAMSAP2-depleted cells with a radial MT array, MTs cannot become enriched in one protrusion, and found that this indeed was the case (*Figure 3G*), strongly suggesting that non-centrosomal MTs stabilize polarized elongated cell morphology by enabling MT enrichment in a single protrusion.

## CAMSAP2 depletion phenotypes cannot be explained by changes in the actin cytoskeleton and cell contractility

The reduced protrusion persistence prompted us to examine the organization of acto-myosin cytoskeleton after CAMSAP2 silencing. In 2D-cultured CAMSAP2-depleted cells, we observed a modest increase in the density of F-actin cytoskeleton (*Figure 4A*), due to the presence of more stress fibers (*Figure 4—figure supplement 1A*). Beside this small difference, cells established normal polarized front-rear morphologies, as revealed by the presence of actin-enriched lamellipodia and myosin IIb-positive retracting cell edges, and the distribution patterns of active Rho and Rac1, as well as cell adhesion markers were normal (*Figure 4A,B*, *Figure 4—figure supplement 1A,B,C*). Importantly, actin cytoskeleton was unchanged in 3D environment in the absence of CAMSAP2, displaying intense peripheral cortical accumulation as in control situation (*Figure 4C*, *Figure 4—figure supplement 1D*).

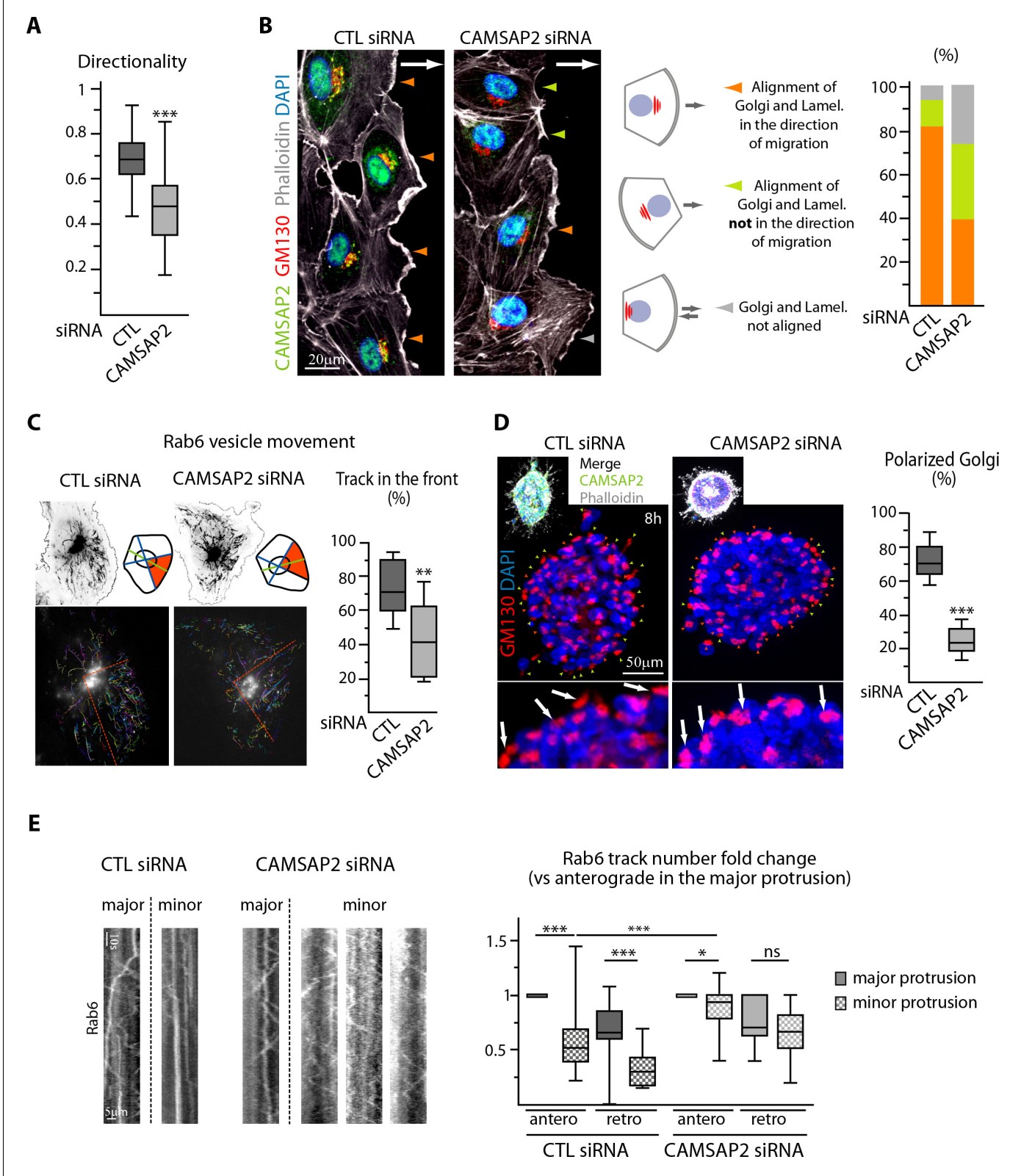

**Figure 5.** CAMSAP2 participates in Golgi polarization during 2D migration and 3D sprouting. (**A**) Directionality of cell movement (distance between the start and end point of migration divided by the total distance travelled) during a phase-contrast time-lapse recording of a wound healing assay after CAMSAP2 knockdown, n = 60 cells in two independent experiments per condition. (**B**) HUVECs during 2D wound healing assay stained for CAMSAP2 (green), Golgi (GM130, red), F-actin (phalloidin, white) and DNA (DAPI, blue). Z-maximum projections of confocal images are shown; the arrow points
*Figure 5 continued on next page*

*Figure 5 continued*

toward the wound. Colored arrowheads show distinct cell behaviors quantified in the plot on the right, n = 70 and 79 cells in two independent experiments. Orange arrowhead, lamellipodia and the Golgi face the wound, green arrowhead, lamellipodia do not face the wound but align with the Golgi, grey arrowhead, lamellipodia and Golgi not aligned. (C) Rab6 vesicle tracks in HUVECs transfected with control or CAMSAP2 siRNA. Fluorescence time-lapse TIRF images of GFP-Rab6A at the front cell row in a wound healing assay were tracked automatically. Maximum intensity projections of the acquired signal (black, top) and tracks resulting from automatic processing (multicolor, bottom) are shown. The front of the cell was defined according to the front-rear cell morphology (the orange area in the scheme and the dashed orange line in the bottom images), and the proportion of tracks in this area was quantified, n = 12 cells per condition. (D) Staining for Golgi (GM130, red) and DNA (DAPI, blue) in control or CAMSAP2-depleted HUVEC spheroids 8 hr after 3D spheroid embedding. Upper images display additional CAMSAP2 (green) and F-actin (phalloidin, white) staining. Z-maximum projections of confocal images are shown; in the zoomed images, arrows point to the Golgi positioned in the direction of sprouting in control cells and behind the nuclei in CAMSAP2-depleted cells. Plot shows the proportion of cells at the spheroid periphery with polarized Golgi (defined as having their Golgi jutting out more than 50% of their surface area in front of the nucleus); the green and orange arrowheads point to cells containing polarized or non-polarized Golgi, respectively; n = 10 spheroids per condition. (E) Kymographs illustrating transport of exocytotic vesicles labeled with GFP-Rab6A in the major (longest) and the other (minor) protrusions in control or CAMSAP2-depleted HUVECs. The number of anterograde and retrograde tracks in the major and minor protrusion(s) was quantified for each condition and normalized by the anterograde track number in the major protrusion (plot), n = 20 and 17 cells in two independent experiments. Data are shown using box plots; Mann-Whitney U test (: ***p<0.001, **p<0.01, *p<0,05, ns, no significant difference.

DOI: https://doi.org/10.7554/eLife.33864.025

The following source data and figure supplements are available for figure 5:

**Source data 1.** An Excel sheet with numerical data on the quantification of the effect of CAMSAP2 depletion on the directionality of EC migration, the correlation between the position of the lamellipodia, Golgi and wound during migration, the proportion of Rab6 tracks in the front of migrating ECs, the polarization of Golgi in sprouting ECs and the proportion of Rab6 tracks (anterograde and retrograde) in the 3D longest protrusion represented as plots in *Figure 5A–E*.

DOI: https://doi.org/10.7554/eLife.33864.027

**Figure supplement 1.** CAMSAP2 participates in Golgi polarization during 2D migration and 3D sprouting.

DOI: https://doi.org/10.7554/eLife.33864.026

**Figure supplement 1—source data 1.** An Excel sheet with numerical data on the quantification of the effect of CAMSAP2 depletion on the polarization of the centrosome during migration and the speed, duration and length of Rab6 tracks in 2D migrating ECs represented as plots in *Figure 5—figure supplement 1A,B*.

DOI: https://doi.org/10.7554/eLife.33864.028

Myosin II-dependent cell contractility has been shown to inhibit protrusion formation in ECs (*Fischer et al., 2009*). In agreement with these observations, inhibition of myosin II either directly, with blebbistatin, or indirectly, with the inhibitor of the kinase ROCK (Y-27632), led to longer 3D protrusions in each condition (*Figure 4—figure supplement 2A–D*). However, in the absence of CAMSAP2, these treatments did not rescue the polarized elongated morphology typical of control ECs (*Figure 4D*, *Figure 4—figure supplement 2A,C,E*). Moreover, although decreased contractility facilitated protrusion persistence, CAMSAP2 depletion still severely reduced protrusion stability in the presence of the ROCK inhibitor (*Figure 4E*). Similarly, treatment of spheroids with the ROCK inhibitor or blebbistatin induced more and longer sprouts but failed to rescue sprouting impairment in CAMSAP2-depleted cells (*Figure 4F*, *Figure 4—figure supplement 2F*). Altogether, these data demonstrate that changes in acto-myosin cytoskeleton or contractility cannot explain the defects associated with CAMSAP2 silencing.

## CAMSAP2 participates in Golgi polarization during 2D migration and 3D sprouting

While displaying normal front-rear morphologies, CAMSAP2-depleted cells were often unable to orient their lamellae in the direction of migration, suggesting polarity defects (*Figure 4B*, asterisks). Accordingly, CAMSAP2 inactivation resulted in a substantial drop in the directionality of cell movement (*Figure 5A*), which was associated with Golgi and centrosome mispositioning respective to the wound (*Figure 5B*, *Figure 5—figure supplement 1A*). This explained the global impairment of cell migration in spite of only a minor decrease in movement velocity (*Figure 2—figure supplement 1I*). Furthermore, the correlation between the positions of the Golgi and the centrosome, which always colocalized (*Figure 5—figure supplement 1A*), and the lamellipodia was strongly diminished after CAMSAP2 silencing, with some ECs having their leading edge in front of the Golgi and some not (*Figure 5B*). Forward-targeted post-Golgi vesicle trafficking, which is often considered as a key

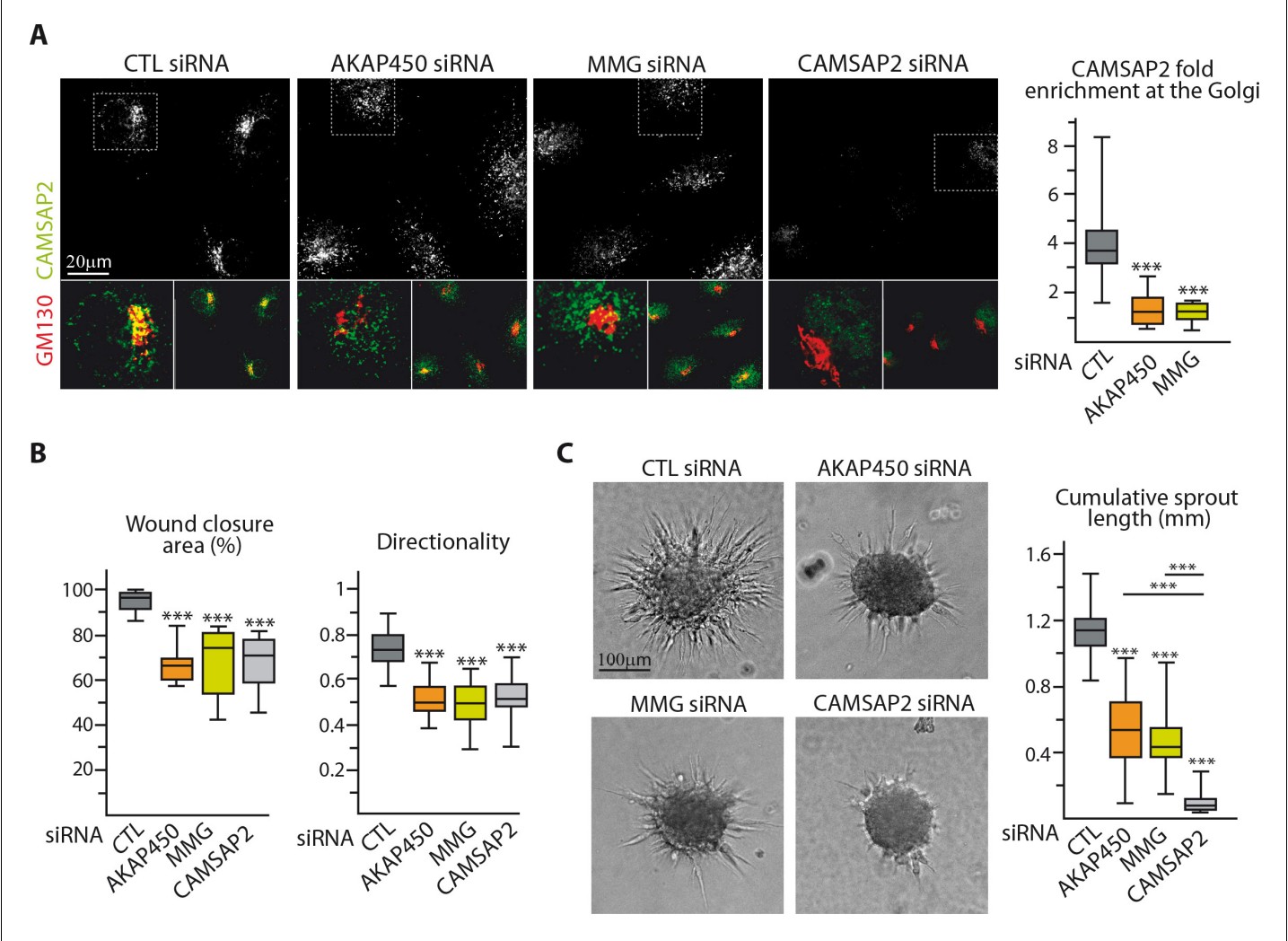

**Figure 6.** Loss of non-centrosomal MTs has a more severe impact than their detachment from the Golgi in 3D but not in 2D. (**A**) Staining for CAMSAP2 (white, green) and Golgi (GM130, red) in HUVECs transfected with the indicated siRNA. The plot shows CAMSAP2 enrichment at the Golgi (ratio between the average CAMSAP2 intensity on top of Golgi and in the cytoplasm), n = 20 cells per condition. (**B**) Quantification of migratory parameters during wound healing assay in HUVECs transfected with the indicated siRNA. Bright-field micrographs were taken before and 8 hr after wounding to calculate the percentage of wound closure, n = 16 fields in two independent experiments per condition. Cell tracking analysis was performed on phase-contrast live imaging to quantify the directionality of cell movement, n = 30 cells per condition. (**C**) Spheroid sprouting assay in HUVECs transfected with the indicated siRNAs. Plots of the cumulative length of all sprouts per spheroid are on the right, n = 58, 62, 63 and 58 spheroids in three independent experiments. Data are shown using box plots; Mann-Whitney U test: ***p<0.001.

DOI: https://doi.org/10.7554/eLife.33864.029

The following source data and figure supplements are available for figure 6:

**Source data 1.** An Excel sheet with numerical data on the quantification of the enrichment of CAMSAP2 at the Golgi, the directionality and efficiency of migration during scratch-wound assays and the cumulative length of spheroid sprouts in the absence of AKAP450, MMG or CAMSAP2 represented as plots in *Figure 6A–C*.

DOI: https://doi.org/10.7554/eLife.33864.031

**Figure supplement 1.** Loss of non-centrosomal MTs has a more severe impact than their detachment from the Golgi in 3D but not in 2D.

DOI: https://doi.org/10.7554/eLife.33864.030

**Figure supplement 1—source data 1.** An Excel sheet with numerical data on the quantification of the profile of CAMSAP2 and GM130 signal along the cell radius, the velocity of migration and the polarization of Golgi during scratch-wound assays in the absence of AKAP450, MMG or CAMSAP2 represented as plots in *Figure 6—figure supplement 1B–D*.

DOI: https://doi.org/10.7554/eLife.33864.032

regulator of cell polarity during migration, was perturbed in the absence of CAMSAP2. Whereas in control ECs the majority of exocytotic vesicles labeled with the small GTPase Rab6 moved towards the leading edge, Rab6 trajectories were distributed more symmetrically in CAMSAP2-depleted ECs, while the other trafficking parameters were not affected (*Figure 5C*, *Figure 5—figure supplement 1B*). This likely reflected a more radial, symmetric MT array in these cells (*Figure 2C*, *Figure 2—figure supplement 1D*). Together, our 2D results argue in favor of a model where Golgi positioning does not directly dictate the orientation of the extending lamellae but is needed to stabilize directional persistence by controlling polarized trafficking.

Forward positioning of the Golgi, which strongly co-localized with CAMSAP2 stretches, was even more striking in 3D (*Figure 5—figure supplement 1C*). In this 3D setting, CAMSAP2 proved to be essential for Golgi polarization in the direction of sprouting (*Figure 5D*). In line with our 2D observations, while control cells displayed highly asymmetric trafficking of MT-dependent Rab6 positive exocytotic vesicles, little asymmetry in trafficking was observed between protrusions of CAMSAP2-depleted cells (*Figure 5E*). Consistent with the inability of centrosome-anchored MTs to redistribute to a single protrusion (*Figure 3G*), the failure in polarization of the secretory trafficking, which is likely necessary to generate a stable leading cell edge (*Schmoranzer et al., 2003*; *Stehbens et al., 2014*; *Yadav et al., 2009*) could explain the inability of CAMSAP2 depleted ECs to form mature sprouts in 3D.

## Loss of non-centrosomal MTs has a more severe impact than their detachment from the Golgi in 3D but not in 2D

The above results suggest that by protecting Golgi-tethered MTs, CAMSAP2 regulates proper Golgi positioning important for EC polarization. To test this idea, we made use of our recent findings showing that two proteins, AKAP450 and myomegalin (MMG), are needed for anchoring CAMSAP2-decorated MT minus-ends to Golgi membranes (*Wu et al., 2016*). As expected, depletion of AKAP450 or MMG (*Figure 6—figure supplement 1A*) displaced CAMSAP2 stretches from the Golgi (*Figure 6A*, *Figure 6—figure supplement 1B*). This redistribution had remarkably similar consequences for 2D migration compared to CAMSAP2 silencing: ECs that had lost their Golgi-attached CAMSAP2 were unable to maintain directionality and to orient their Golgi, resulting in migration deficiency (*Figure 6B*, *Figure 6—figure supplement 1C,D*). Interestingly, the situation differed when ECs were placed in a 3D context: whereas displacing non-centrosomal CAMSAP2-bound MT ends from Golgi by depleting AKAP450 or MMG had a negative impact on endothelial sprouting abilities, its effect was significantly milder than that of CAMSAP2 depletion, when most MTs were attached to the centrosome (*Figure 6C*). This suggests that in 3D sprouting, non-centrosomal MTs might have a role independent of their direct Golgi association.

## Centrosome removal promotes cell polarization in the absence of CAMSAP2

We reasoned that non-centrosomal MTs, which are not anchored to a single point, might redistribute more easily to create asymmetry, and therefore, centrosome removal in CAMSAP2-depleted ECs might improve their polarization potential by restoring a pool of non-centrosomal MTs. To test this idea, we silenced AKAP450, MMG and CAMSAP2 in ECs in combination with centrinone-induced centrosome depletion. Such ECs were viable and efficiently lost their centrosome, displaying a characteristic enlarged shape filled with a dense non-centrosomal MT array, even in the absence of CAMSAP2 (*Figure 7A*). The density of MTs and growing, EB3-positive MT ends was similar in all conditions (*Figure 7B*, *Figure 7—figure supplement 1A*). As we described in our previous study (*Wu et al., 2016*), acentrosomal cells showed enhanced recruitment of γ-tubulin to the Golgi, an effect that was abolished by depleting AKAP450 (*Figure 7C,D*). In line with these results, we observed abundant MT nucleation from the Golgi membranes after nocodazole washout in control, MMG- and CAMSAP2-depleted acentrosomal cells, while in AKAP450-depleted acentrosomal cells, MTs were nucleated from the cytoplasmic sites distinct from the Golgi membranes (*Figure 7—figure supplement 1B*). Such distribution of MT nucleation sites correlated with the recruitment of the pericentriolar material (PCM) marker Pericentrin to the Golgi in control, MMG- and CAMSAP2-depleted centrinone-treated cells, and its dispersion in the cytoplasm in AKAP450-depleted cells (*Figure 7—figure supplement 1C*).

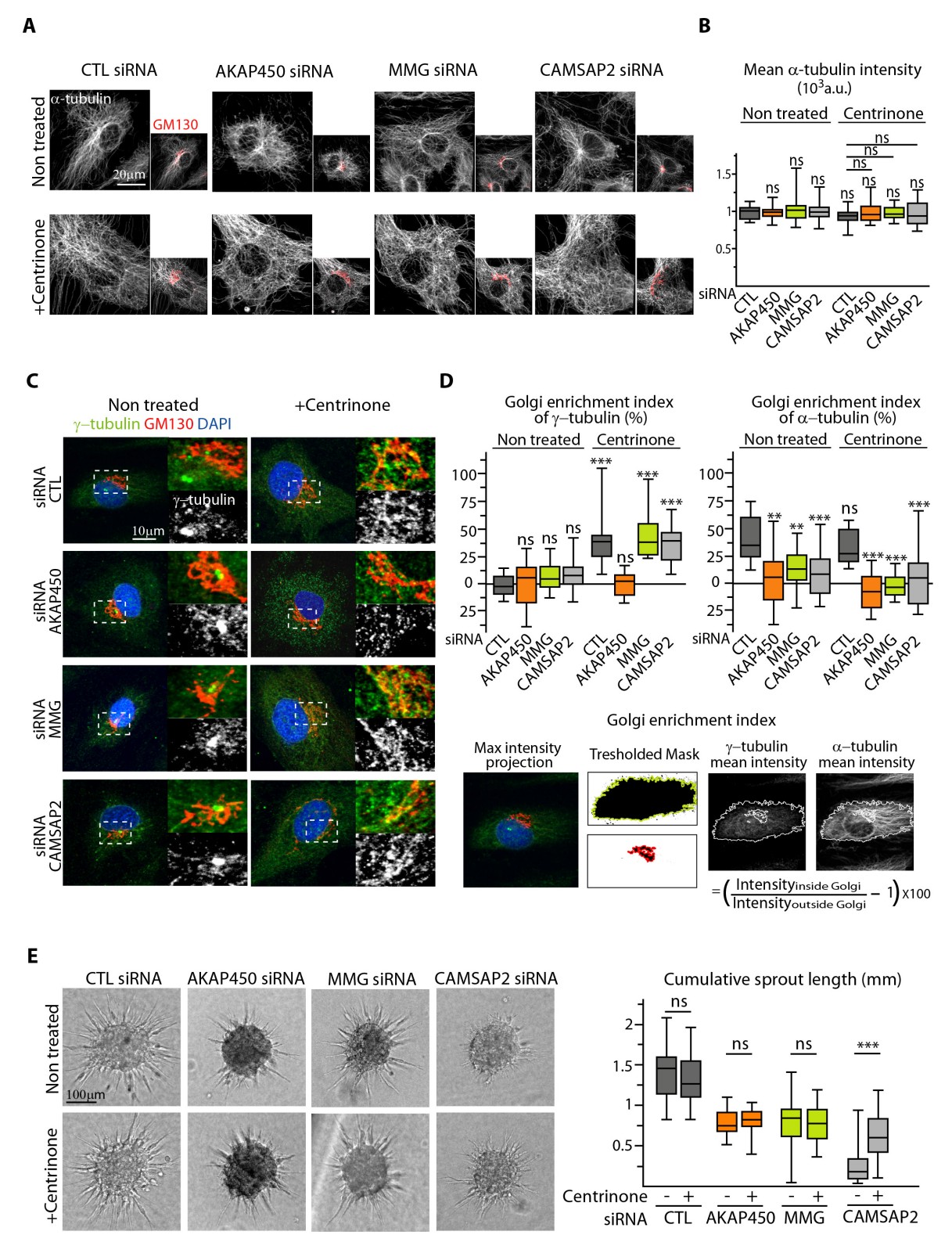

**Figure 7.** The centrosome inhibits cell polarization and sprouting in the absence of CAMSAP. (**A,B**) Staining for Golgi (GM130, red) and α-tubulin (white) in control and centrinone-treated HUVECs transfected with the indicated siRNAs. Z-maximum projections of confocal images (**A**) and average fluorescence intensity of α-tubulin (**B**) are shown; n = 25 cells in each condition. Histogram bars for non-treated/control siRNA, non-treated/CAMSAP2 siRNA and centrinone treated/control siRNA are the same as in *Figure 1A* and *Figure 2—figure supplement 1C*. (**C**) Staining for centrosome (γ-

*Figure 7 continued on next page*

*Figure 7 continued*

tubulin, green/white), Golgi (GM130, red) and DNA (DAPI, blue) in control and centrinone-treated HUVECs transfected with the indicated siRNAs. Z-maximum projections of confocal images are shown. (D) Enrichment index (difference between the average intensity at the Golgi and in the rest of the cell, divided by the intensity in the cytoplasm expressed in percent) was quantified as schematized at the bottom and as described in Materials and methods for γ-tubulin and α-tubulin at the Golgi in HUVECs treated and stained as in (A) and (C), n = 16, 12, 13, 14, 33, 20, 14 and 15 cells (γ-tubulin) and n = 15, 14, 13, 14, 14, 16, 14 and 15 cells (α-tubulin). (E) Spheroid sprouting assay of HUVECs in the indicated conditions in presence of thymidine. Plot shows quantification of the cumulative length of all sprouts per spheroid, n = 61, 59, 33, 35, 42, 39, 53 and 65 spheroids in three independent experiments. Data are shown using box plots; Mann-Whitney U test: **p<0.01, ***p<0.001, ns, no significant difference.
DOI: https://doi.org/10.7554/eLife.33864.033

The following source data and figure supplements are available for figure 7:

**Source data 1.** An Excel sheet with numerical data on the quantification of the mean intensity of EC α-tubulin signal, the enrichment of γ- and α-tubulin at the Golgi and the cumulative length of spheroid sprouts in the absence of AKAP450, MMG or CAMSAP2 and after centrinone treatment represented as plots in *Figure 7B,D,E*.
DOI: https://doi.org/10.7554/eLife.33864.035

**Figure supplement 1.** The centrosome inhibits cell polarization and sprouting in the absence of CAMSAP2.
DOI: https://doi.org/10.7554/eLife.33864.034

**Figure supplement 1—source data 1.** An Excel sheet with numerical data on the quantification of the number of EB comets in 2D ECs and the enrichment of EB at the Golgi after nocodazole washout in the absence of AKAP450, MMG or CAMSAP2 together with centrinone treatment represented as plots in *Figure 7—figure supplement 1A,B*.
DOI: https://doi.org/10.7554/eLife.33864.036

In agreement with our previous work in RPE1 cells (*Wu et al., 2016*), MT density was detached from the Golgi region after AKAP450 and MMG knockdown, because these proteins constitute a part of an essential link between MTs and the Golgi membranes (*Figure 7A,D*). In CAMSAP2-depleted ECs, centrinone treatment caused some disorganization of the Golgi, and MTs were not concentrated in the Golgi area either (*Figure 7A,D*).

The absence of the centrosome caused no additional reduction in the sprouting ability of AKAP450 or MMG-depleted ECs organized in spheroids (*Figure 7E*), indicating that CAMSAP2-stabilized non-centrosomal MTs are sufficient to support formation of elongated sprouts from spheroids to some extent even when they are not attached to the Golgi. Strikingly, the removal of centrosome in the absence of CAMSAP2 significantly rescued the sprouting potential of ECs (*Figure 7E*). These data support the idea that non-centrosomal MTs contribute positively to EC sprouting, while the centrosome is not only dispensable, but can also play an inhibitory role when CAMSAP2 is absent.

## Non-centrosomal MTs are required to create protrusion asymmetry

Among all the situations analyzed, CAMSAP2-depleted cells which had centrosomes were the only ones which had a symmetric, strongly radial MT system, while this property was lost when these cells were treated with centrinone (*Figure 7A*, *Figure 8A*). As described above, CAMSAP2-depleted cells with centrosomes had symmetric radial protrusions in 3D (*Figure 3B–D*), but, remarkably, centrosome depletion restored their ability to generate one long dominant protrusion (*Figure 8B*, *Figure 8—figure supplement 1*). MMG-depleted cells, in which non-centrosomal MTs are present but not anchored at the Golgi performed in these assays just as well as control cells irrespective of their centrosome content (*Figure 8B*), and we found that although CAMSAP2-decorated minus ends were not enriched at the Golgi anymore after MMG depletion, they were still concentrated in the major protrusion (*Figure 8C*). Therefore, Golgi attachment is not a requirement for concentrating non-centrosomal MTs in one cell protrusion in 3D. This observation likely explains why centrosome removal rescues sprouting in CAMSAP2-delpeted cells: non-centrosomal MTs can concentrate in one protrusion (*Figure 8D*) even though they are not linked to Golgi membranes. Taken together, our data demonstrate that the presence of non-centrosomal MTs is essential for generating cell asymmetry required for the emergence of long EC sprouts, while attachment of these MTs to the Golgi, and the likely more efficient secretion associated with such an arrangement is beneficial but not essential.

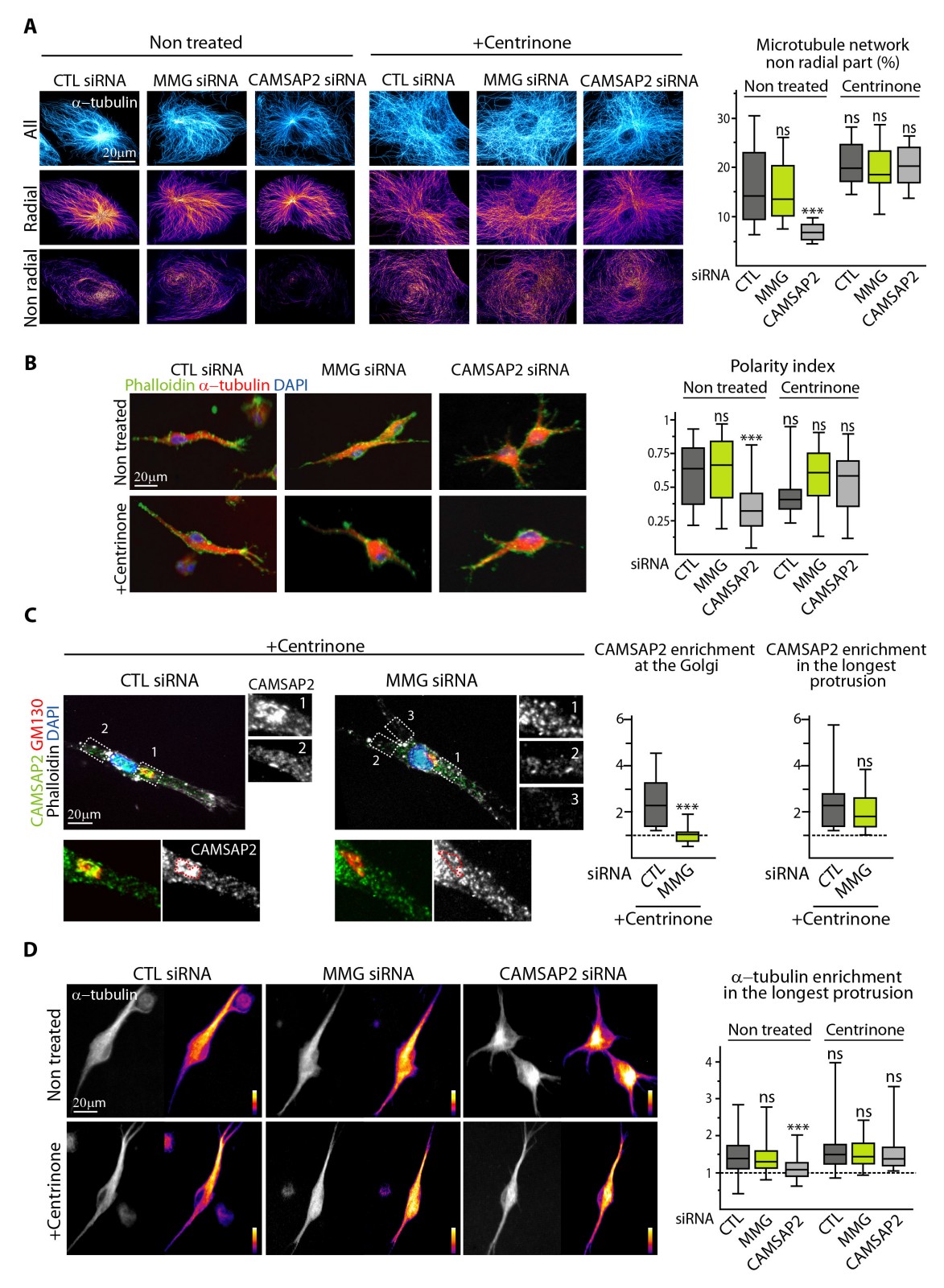

**Figure 8.** Non-centrosomal MTs are required to create protrusion asymmetry. (**A**) Imaging of control and centrinone-treated HUVECs transfected with the indicated siRNAs and stained for MTs (α-tubulin, cyan hot) using STED microscopy. MT images were split into a radial and non-radial component based on MT orientation in relation to the centrosome or the brightest point as described in the Materials and methods. The resulting heat maps (shown using a color look-up table (LUT)) were used to quantify the proportion of the non-radial part of the MT network in a circular section around the

*Figure 8 continued on next page*

*Figure 8 continued*

reference point (see Materials and methods for details), n = 12 cells per condition. (**B**) Staining for F-actin (phalloidin, green), α-tubulin (red) and DNA (DAPI, blue) in 3D cultured HUVECs treated as in (**A**). Z-maximum projections of confocal fluorescence images were used to calculate polarity index as described in Materials and methods and the legend to *Figure 3D*, n = 44, 36, 35, 38, 36 and 39 cells. (**C**) Staining for CAMSAP2 (green, white), Golgi (GM130,red), F-actin (phalloidin, white) and DNA (DAPI, blue) in 3D cultured HUVECs treated with centrinone and transfected with control or MMG siRNA. Z-maximum projections of confocal fluorescence images are shown and were used to calculate CAMSAP2 signal intensity enrichment at the Golgi (ratio between the average signal in the Golgi area and in the cytoplasm) and in the longest protrusion (ratio between the average signal in the longest protrusion (box 1) and in the other protrusions), n = 16 and 18 cells. (**D**) Staining for α-tubulin in 3D cultured HUVECs treated as in (**A**). Z-maximum projections of confocal fluorescence images are shown using a grey (left) or a color look-up table (LUT) (right) and were used to calculate the average α-tubulin intensity ratio between the longest protrusion and the other ones, n = 36 cells for each conditions. Data are shown using box plots; Mann-Whitney U test: \*\*\*p<0.001, ns, no significant difference.

DOI: https://doi.org/10.7554/eLife.33864.037

The following source data and figure supplements are available for figure 8:

**Source data 1.** An Excel sheet with numerical data on the quantification of the effect of MMG or CAMSAP2 depletion together with centrinone treatment on the proportion of the non-radial MT network, the EC polarity index and the enrichment of α-tubulin signal in the 3D longest protrusion as well as the effect of MMG depletion and centrinone treatment on the enrichment of CAMSAP2 at the Golgi and in the longest protrusion in 3D represented as plots in *Figure 8A–D*.

DOI: https://doi.org/10.7554/eLife.33864.039

**Figure supplement 1.** Non-centrosomal MTs are required to create protrusion asymmetry.

DOI: https://doi.org/10.7554/eLife.33864.038

**Figure supplement 1—source data 1.** An Excel sheet with numerical data on the quantification of the effect of MMG or CAMSAP2 depletion together with centrinone treatment on the number and length of the 3D protrusions of isolated ECs represented as plot in *Figure 8—figure supplement 1*.

DOI: https://doi.org/10.7554/eLife.33864.040

## CAMSAP2 plays a role in sprouting angiogenesis in vivo

Finally, we addressed the role of CAMSAP2 in vivo using zebrafish as a model. Two CAMSAP2-encoding gene orthologues are present in zebrafish, *camsap2a* (ENSDARG00000062173) and *camsap2b* (ENSDARG00000059965). To analyze the role of CAMSAP2 in zebrafish vascular development, we designed splice-blocking antisense morpholinos (*Figure 9—figure supplement 1A*) to generate CAMSAP2-silenced embryos in the endothelial reporter line *Tg(fli1a:eGFP)*. Embryos inactivated for CAMSAP2a or CAMSAP2b were viable, had no obvious morphological defects and normal somite development. We next focused on CAMSAP2b, because among CAMSAP2 orthologs, it showed the highest expression in the zebrafish endothelium and was associated with more severe vascular defects. In zebrafish, two waves of dorsal sprouting angiogenesis take place successively during vascular development (*Ellertsdóttir et al., 2010*; *Isogai et al., 2003*). The first one occurs at around 22 hr post fertilization (hpf) from the dorsal aorta and forms arterial intersegmental vessels. Another one takes place between 32 and 48 hpf from the cardinal vein and gives rise to venous intersegmental vessels and to parachordal lymphangioblasts, precursors of lymphangiogenic vessels (*Figure 9A*). After CAMSAP2b inactivation (*Figure 9—figure supplement 1B*), arterial intersegmental vessel formation was hardly altered, but the secondary EC sprouting was perturbed, giving rise to abnormal tortuous venous intersegmental vessels (*Figure 9—figure supplement 1C*). In CAMSAP2b morphants, we observed fewer secondary sprouts emerging from the cardinal vein at 34 hpf (*Figure 9—figure supplement 1D*). In addition, the fraction of venous sprouts that had fused with the neighboring arterial intersegmental vessel at 41 hpf was strongly reduced (*Figure 9—figure supplement 1D*), suggesting a defect in directional migration. Venous intersegmental vessels form after cardinal vein-derived secondary sprouts connect to primary arterial intersegmental vessels. Because secondary sprouts had difficulties to fuse with primary intersegmental vessels, we also found that the proportion of venous intersegmental vessels at 48hpf was reduced in CAMSAP2b morphants: whereas control embryos displayed a typical 50–50 arterial/venous intersegmental vessel ratio, fewer intersegmental vessels were connected to the cardinal veinand scored as venous intersegmental vessels in CAMSAP2b morphant embryos (*Figure 9A,B*). In addition, the alternative outcome of venous sprouting, that is parachordal lymphangioblasts formation, was alsoimpaired in the absence of CAMSAP2b, with less or aberrant parachordal lymphangioblasts in the morphants (*Figure 9A*, *Figure 9—figure supplement 1E*, asterisks). Importantly, injection of mRNA coding for human CAMSAP2

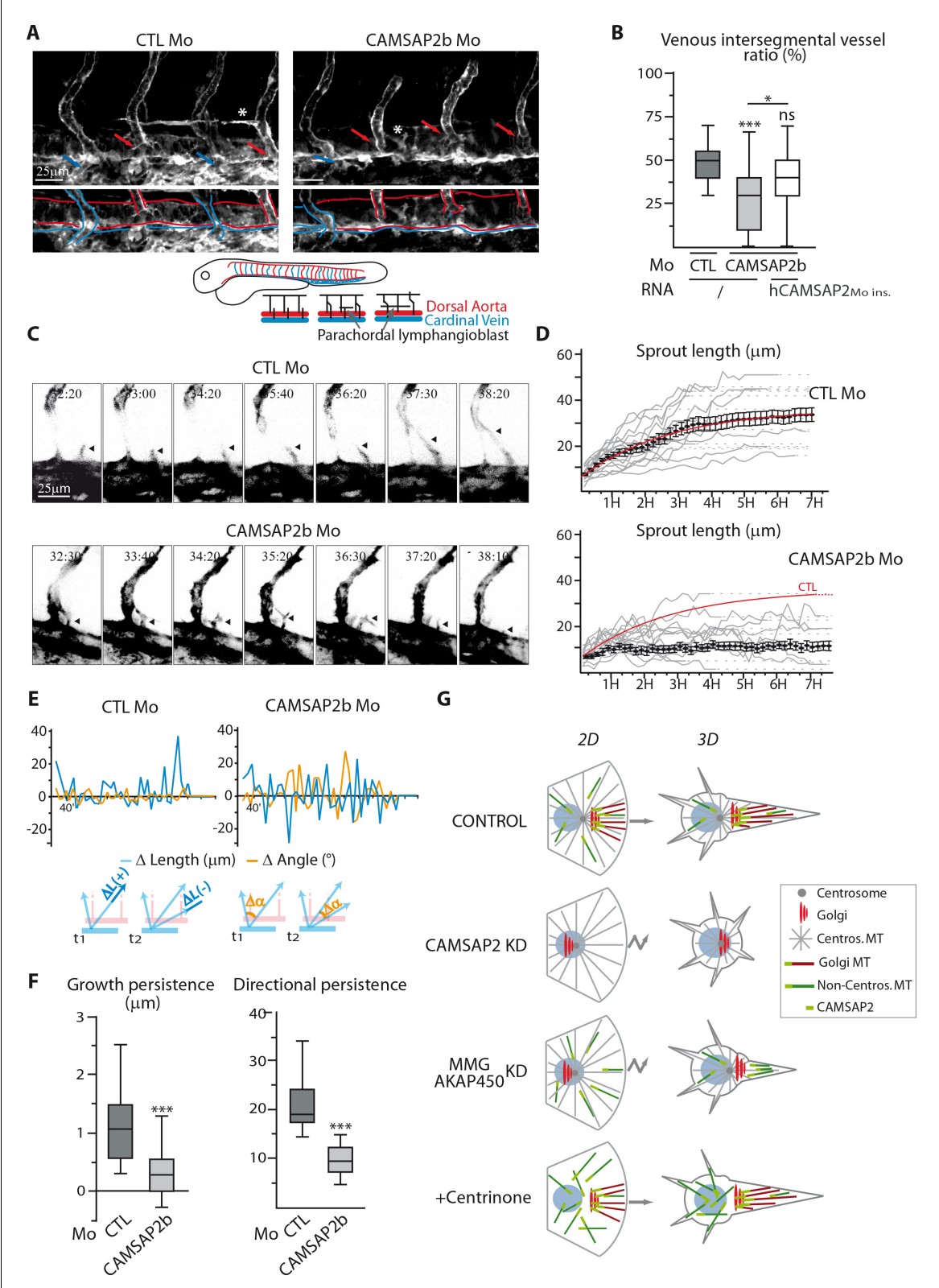

**Figure 9.** CAMSAP2 plays a role in sprouting angiogenesis in vivo. (**A**) Live confocal images (Z-maximum projections) of 48 hpf *Tg(Fli1ep:Lifeact-EGFP)* embryos injected with control or CAMSAP2b morpholinos. Arterial and venous intersegmental vessels are indicated by red and blue arrows and highlighted with red and blue lines on the bottom pictures, respectively. Asterisks show parachordal lymphangioblast in control embryo and an abnormal venous sprout in CAMSAP2b morphant embryo. (**B**) Quantification of the percentage of venous intersegmental vessels in the same 10 somite-

*Figure 9 continued on next page*

*Figure 9 continued*

region in the trunk of embryos injected with control or CAMSAP2b morpholinos, or co-injected with CAMSAP2b morpholinos and RNA coding for a morpholino-insensitive mutant of human CAMSAP2, n = 90, 80 and 45 embryos in six, six and three independent experiments. (**C**) Frames (Z-maximum projections) from time-lapse confocal imaging of venous sprouting in control and CAMSAP2b-depleted embryos. Time is hr:min post-fertilization. Arrowheads point to the growing venous sprout. See also *Videos 1–5*. (**D**) Graphs representing venous sprout length over time measured during their growing period from the time-lapse imaging described in (**C**) in control and CAMSAP2b morphant embryos. Grey curves represent individual growing events, black dots indicate the average length at each time point ±SEM and the result of curve fitting (exponential - one phase association) in control embryos is drawn in red. (**E**) Graphs representing length (blue lines, ΔL in the scheme) and angle (orange lines, Δα in the scheme) variations of growing venous sprouts between each successive time point (t1 and t2 in the scheme) from the time-lapse imaging described in (**C**). One representative plot (out of 19 and 18) is shown for each condition. (**F**) Quantification of the average growth and directional persistence per growing event calculated from data described in (**E**) and as explained in the Materials and methods: the growth persistence was obtained by averaging the length variations (Δ Length) between two consecutive time frames per growing event whereas directional persistence was calculated as the inverse of the sinus of the angle variation (its absolute value) for each frame and then averaged per growth event, n = 19 and 18 sprouts in three independent experiments. (**G**) Model of the impact of MT array organization on endothelial polarization and movement in 2D and 3D. Control ECs contain three distinct populations of MT, the centrosomal MTs (grey), the non-centrosomal, Golgi-anchored MTs (burgundy) and the non-centrosomal non-Golgi-anchored MTs (dark green). The two non-centrosomal MT populations are stabilized by the presence of CAMSAP2 stretches at their minus-ends (light green rectangle). During 2D migration, the presence of Golgi-originating MTs, which are lost after CAMSAP2, MMG or AKAP450 depletion, ensures proper Golgi polarization and directional migration. In the context of 3D sprouting, both non-centrosomal populations are enriched in a single protrusion, which becomes larger and more stable than the rest. Centrosomal MTs are dispensable for both processes. Data are shown using box plots except in (**D**): mean ±SEM; Chi square test with Yates correction (**B**), Mann-Whitney U test (**F**): ***$p<0.001$, *$p<0.05$, ns, no significant difference.

DOI: https://doi.org/10.7554/eLife.33864.041

The following source data and figure supplements are available for figure 9:

**Source data 1.** An Excel sheet with numerical data on the quantification of the effect of CAMSAP2b inactivation and its re-expression (for 1B) in zebrafish on the proportion of venous intersegmental vessels, the length of venous sprouts over time, the variation of their length and direction over time and the growth and directional persistence of secondary sprout formation represented as plots in *Figure 9B,D–F*.
DOI: https://doi.org/10.7554/eLife.33864.043

**Figure supplement 1.** CAMSAP2 plays a role in sprouting angiogenesis in vivo.
DOI: https://doi.org/10.7554/eLife.33864.042

**Figure supplement 1—source data 1.** An Excel sheet with numerical data on the quantification of the efficiency of CAMSAP2b silencing, the number of secondary venous sprouts at 34 and 48 hpf and the number of loops in the caudal vein plexus after CAMSAP2b-directed morpholino injection in zebrafish embryos as well as of the number of secondary venous sprouts at 36 hpf after re-expression of CAMSAP2 represented as plots in *Figure 9—figure supplement 1B,D,F,G*.
DOI: https://doi.org/10.7554/eLife.33864.044

restored normal phenotypes in the majority of the morphants (*Figure 9B*, *Figure 9—figure supplement 1F*).

To confirm these results, we performed live imaging of venous sprouting in *Tg(Fli1ep:Lifeact-EGFP)*, a zebrafish line with F-actin labeled in the endothelium (*Phng et al., 2013*). In control conditions, venous ECs migrated in a highly directed manner to either fuse to the neighboring arterial intersegmental vessel or start assembling horizontal parachordal lymphangioblasts (Supplemental *Videos 1* and *2*, *Figure 9C*). In contrast, in CAMSAP2b morphants venous sprouts were very unstable and showed less directional persistence (Supplemental *Videos 3* and *4*, *Figure 9C*), sometimes resulting in the atypical fusion of two distinct sprouts with the same arterial intersegmental vessel (Supplemental *Video 5*). These observations were quantitatively validated by tracking the length and orientation of venous sprouts over time. In contrast to their regular extension in control animals, venous sprouts depleted of CAMSAP2b displayed a more erratic and less efficient growth (*Figure 9D*). This aberrant behavior is also illustrated in *Figure 9E*, where CAMSAP2b silencing is shown to be associated with a lot of shortening episodes (negative change in length) and a more variable sprout orientation (higher change in angle), resulting in a significantly lower growth and directional persistence (*Figure 9F*). Supporting a role for CAMSAP2 in venous angiogenesis, formation of the caudal vein plexus, a honeycomb-like structure arising from active ventral migration of venous ECs from the cardinal vein in the caudal region, was also perturbed in the morphants (*Figure 9—figure supplement 1G*). Altogether, these observations suggest a defect in guided migration during venous sprouting in CAMSAP2b-silenced zebrafish embryos, in agreement with our in vitro findings and supporting the idea that CAMSAP2 is involved in directional angiogenic sprouting in vivo.

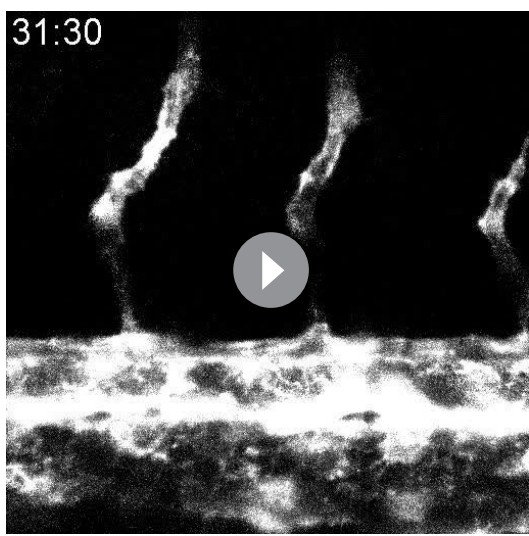

**Video 1.** Time-lapse imaging of directional venous sprouting in control *Tg(Fli1ep:Lifeact-EGFP)* embryos. Time is hr: min post-fertilization. Z-series images in the region centered on the yolk extension end using a 2-μm-step confocal based scan covering 70 μm depth were taken every 10 min. This video shows the highly directional migration of two venous sprouts toward the adjacent arterial intersegmental vessel.
DOI: https://doi.org/10.7554/eLife.33864.045

## Discussion

While the acto-myosin cytoskeleton is crucial for generating protrusions, adhesions and contractile forces during cell migration, an anisotropic MT network strongly contributes to the establishment and maintenance of cell polarity. One current model explaining the generation of an asymmetric MT array involves forward positioning of its center, the centrosome, assumed to represent the main MTOC, together with the local regulation of MT plus end stability (*Etienne-Manneville, 2013*; *Vinogradova et al., 2009*). Here, we showed that although these regulatory processes could contribute to the asymmetry of the system, a centrosomal radial MT network was both completely dispensable and insufficient for the establishment of polarized cell morphology in soft 3D matrices (*Figure 9G*).

Previous work showed that the MT minus-end binding protein CAMSAP2 is a key player in the regulation of non-centrosomal MT minus ends in mammalian cells (*Akhmanova and Hoogenraad, 2015*; *Jiang et al., 2014*; *Tanaka et al., 2012*; *Yau et al., 2014*). Here, we uncovered the crucial role of CAMSAP2 in regulating cell polarity during endothelial sprout formation in 3D and persis-

tent directional migration in 2D (*Figure 9G*). We think that this demonstrates the important role of non-centrosomal MTs in these processes, because, although we cannot exclude that this protein has alternative functions, for example in controlling motor-based transport or signaling, we did not find any direct evidence supporting such functions in our experiments. Furthermore, although MTs are known to regulate actin organization, and MT destabilization promotes myosin II-dependent contractility, which affects protrusion formation (*Etienne-Manneville, 2013*), the protrusions formed in the absence of CAMSAP2 exhibited unchanged F-actin network. This was in line with our observation that the loss of CAMSAP2 had no impact on the overall microtubule density, plus end dynamics or modifications. Our observation that pharmacological inhibition of contractility failed to rescue the persistency and polarized organization of cell protrusions after CAMSAP2 depletion further supports the view that the observed cell morphology and migration defects were not caused by changes in the actin cytoskeleton. Importantly, CAMSAP2-depleted ECs could still form lamellipodia and retracting cell rear on 2D surfaces and initiated protrusions when cultured in 3D. Whether this reflects a sufficient degree of asymmetry supported by the

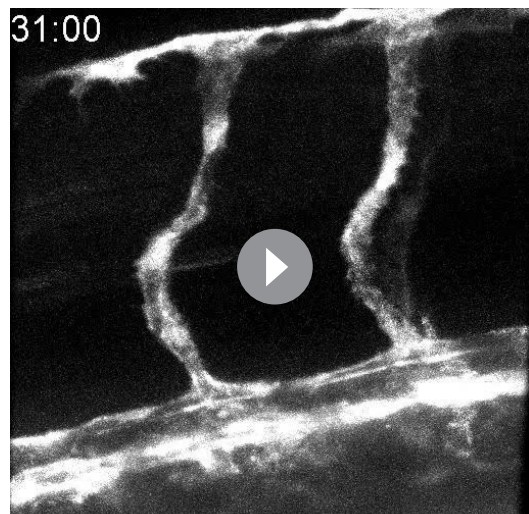

**Video 2.** Time-lapse imaging of venous sprouting, arterial intersegmental vessel fusion and parachordal lymphangioblast assembly in control *Tg(Fli1ep:Lifeact-EGFP)* embryos. Time is hr: min post-fertilization. This video was acquired as described in *Video 1* and illustrates the two different outcomes of venous sprouting: arterial intersegmental vessel fusion or parachordal lymphangioblast assembly, both being directional processes.
DOI: https://doi.org/10.7554/eLife.33864.046

centrosomal MT network, or a lack of MT involvement in these actin-based processes, as it has been suggested in 2D models (*Siegrist and Doe, 2007*), deserves additional investigation. However, both in 2D and 3D, loss of non-centrosomal MTs interfered with persistence of lamellipodia and elongation of cell protrusions, explaining migration defects.

CAMSAP2 played a much more prominent role in 3D than in 2D environment and even displayed different phenotypes in the two distinct 3D settings we used: the absence of CAMSAP2 severely reduced sprouting from spheroids, while the protrusive activity of single isolated ECs in collagen matrix was unaffected. It is likely that the distinct degree of polarity required in these assays could explain this difference. The inability of spheroids to maintain and extend long sprouting structures will culminate in their collapse (as seen in *Figure 3—figure supplement 1A*), whereas isolated ECs could form protrusions in any direction. However, also in the latter case, the radial MT network in CAMSAP2-silenced ECs could not support polarized elongated morphology characteristic for control cells. In our recent study (*Bouchet et al., 2016*), we showed that in

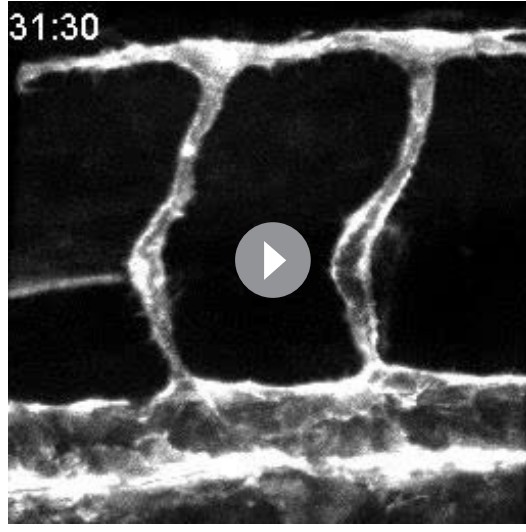

**Video 3.** Time-lapse imaging of unstable venous sprouting in CAMSAP2 morphant *Tg(Fli1ep:Lifeact-EGFP)* embryos. Time is hr: min post-fertilization. This video was acquired as described in *Video 1* and shows two highly unstable venous sprouts.
DOI: https://doi.org/10.7554/eLife.33864.047

mesenchymal cells, the initial formation of protrusions is MT-independent, but the extension and stabilization of long protrusions occurs only if they are filled with MTs. The data presented in the current study indicate that in order to acquire an elongated polarized morphology, 3D-cultured ECs need to be able to concentrate their MTs in one protrusion. If the intrinsic centrosome-driven

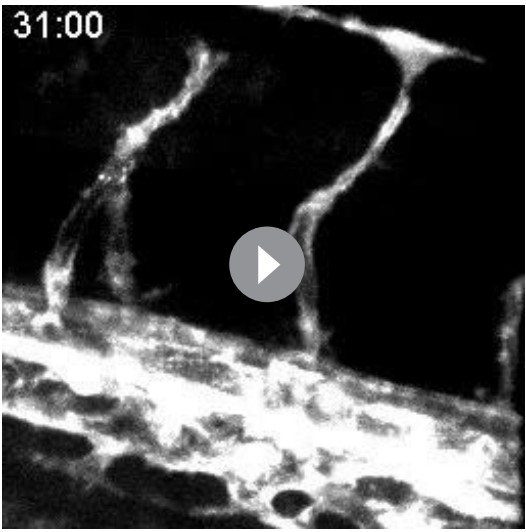

**Video 4.** Time-lapse imaging of non-persistent venous sprouting in CAMSAP2 morphant *Tg(Fli1ep:Lifeact-EGFP)* embryos. Time is hr: min post-fertilization. This video was acquired as described in *Video 1* and illustrates the instability and lack of directional persistence of venous sprouting.
DOI: https://doi.org/10.7554/eLife.33864.048

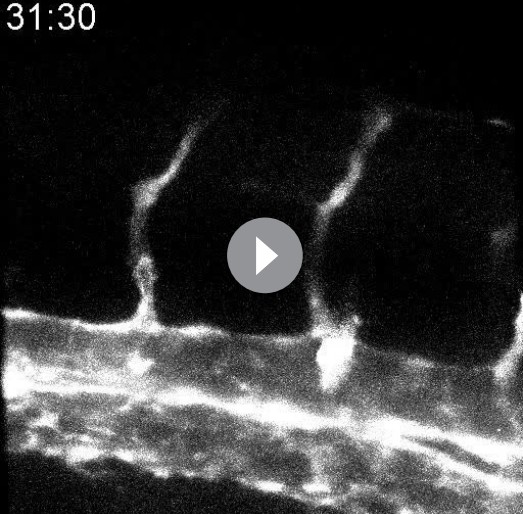

**Video 5.** Time-lapse imaging of venous sprouting in CAMSAP2 morphant *Tg(Fli1ep:Lifeact-EGFP)* embryos. Time is hr: min post-fertilization. This video was acquired as described in *Video 1*. The rightmost arterial intersegmental vessel exhibits atypical fusion with two distinct venous sprouts.
DOI: https://doi.org/10.7554/eLife.33864.049

symmetry of the MT network dominates, it inhibits this process, forcing the cells to acquire a non-polarized 'starfish'-like shape, which is incompatible with efficient cell elongation and translocation in 3D matrix, as observed in the spheroid sprouting assay. While our results focus on one the very first steps of sprouting angiogenesis, that is the extension and stabilization of a protrusion that is required for effective outward migration, other important angiogenic behaviors involving polarity, as multicellular sprout growth and vessel lumenization were not addressed in this study and deserve further investigation.

Non-centrosomal MTs can form intrinsically asymmetric networks through their attachment to the Golgi complex (*Vinogradova et al., 2009*; *Zhu and Kaverina, 2013*). Indeed, confirming previous results (*Roubin et al., 2013*; *Zhu and Kaverina, 2013*), we showed that the depletion of Golgi-originating MTs dependent on AKAP450 and MMG fully recapitulated CAMSAP2 inactivation-related defects in 2D assays. This suggests that the Golgi-anchored MT population, required for the proper Golgi positioning and polarized trafficking, is an important determinant of directional 2D migration. However, in the 3D spheroid assays, the impact of AKAP450 or MMG knockdown was far less severe than that of CAMSAP2 depletion. In fact, an elongated morphology in isolated ECs in 3D could be established when MMG was depleted and Golgi-attached MTs were absent. This suggests that non-centrosomal MTs can function independently of the Golgi anchoring. This idea is supported by the observation that the phenotype caused by the loss of CAMSAP2 could be partially rescued by suppressing formation of a symmetric centrosome-associated MT aster and thus reverting to a non-centrosomal MT array, albeit one lacking Golgi-MT attachments. In fact, the remarkable similarity between the phenotypes associated with CAMSAP2 and MMG depletion after centrinone treatment, leading to similar MT organization, strengthens the idea that the presence of non-centrosomal MTs per se, rather than their Golgi attachment might be crucial to support at least some degree of 3D sprout formation. The redistribution of MT nucleating and anchoring PCM complexes to the Golgi and cytoplasm likely contributes to the generation of such arrays in centrinone-treated cells. It is possible that asymmetrical cortical stabilization of MTs, regulated by extrinsic signals, can be sufficient to polarize the MT network independently of Golgi anchoring, if this polarity is not perturbed by the presence of a potent symmetric MT-anchoring structure such as the centrosome.

We recently showed that a single Golgi apparatus can assemble in the absence of centrosomal and Golgi-derived MTs (*Wu et al., 2016*), suggesting that non-centrosomal MTs that are detached from the Golgi membranes can still regulate Golgi organization. It is therefore possible that non-centrosomal MTs that are not anchored at the Golgi can still control Golgi positioning and ensure polarized secretion, as the Golgi is typically oriented toward the main protrusion in these conditions (*Figure 8C*). In this situation, the Golgi apparatus and non-centrosomal MTs, although not permanently connected, likely exert a positive feedback on each other. The Golgi, which in the absence of the centrosome concentrates γ-tubulin and some other PCM components, serves as the major MT nucleation site, albeit a one which cannot tether the MTs it nucleates. MTs, in turn, serve as directional tracks for localizing Golgi membranes, and increasing MT density in one protrusion will help to maintain the Golgi at the base of this protrusion.

The mechanism of MT minus end stabilization in the absence of both the centrosome and CAMSAP2 is still an open question. An interesting candidate is ninein, a protein involved in MT minus-end organization in epithelial cells (*Moss et al., 2007*), whose worm homolog can act on the same pathway as CAMSAP homologs (*Wang et al., 2015*). Interestingly, ninein has been described to relocalize from the centrosome to the cytoplasm upon triggering angiogenesis and to participate in endothelial morphogenesis (*Matsumoto et al., 2008*).

In line with our in vitro results, CAMSAP2 inactivation in zebrafish perturbed the directional and persistent migration of ECs sprouting dorsally from the cardinal vein. Interestingly, loss of CAMSAP2 had no effect on EC sprouting form the dorsal aorta, thus suggesting that venous sprouting might be more sensitive to the lack of CAMSAP2. Recent evidence suggests that ECs from different vascular beds are differentially regulated and use different mechanisms (*Franco et al., 2016*; *Rocha and Adams, 2009*). It is possible that the secondary wave of sprouting from the cardinal vein is more dependent on intrinsic polarity mechanisms than the arterial sprouting, for instance, due to differential participation of supportive cells and therefore different need for cytoskeleton-based processes.

One of the most surprising findings of this study is the lack of importance of the centrosome in endothelial polarization. Although we do not exclude that centrosomal MTs are participating in polarity establishment in control cells, where the centrosome is in fact the major nucleating factor,

we provide evidence that its MT anchoring activity is dispensable, not sufficient and should be counterbalanced by a non-centrosomal MT population. As it was already described for other organisms (*Bornens, 2012*), our results support the view that the centrosome has no crucial function in many types of animal tissues. Whereas the role of centrosome localization in determining neuronal polarity in vivo has been heavily debated (*Kuijpers and Hoogenraad, 2011*), an inhibitory role for a radial centrosomal MT organization has recently been suggested in epithelia (*Noordstra et al., 2016*). Altogether, our findings support the concept that polarity induction requires a switch to an asymmetric MT network, which might involve participation of centrosome-independent MT minus end stabilizing factors and centrosome inactivation.

# Materials and methods

**Key resources table**

| Reagent type (species) or resource | Designation | Source or reference | Identifiers | Additional information |
|---|---|---|---|---|
| Strain, strain background (*Dario rerio*) | Tg(fli1a:eGFP)y1 | Zebrafish facility GIGA institute, Liege University | ID_Zfin:ZDB-TGCONSTRCT-070117–94 | |
| Strain, strain background (*D. rerio*) | Tg(Fli1ep:Lifeact-EGFP) | Zebrafish facility GIGA institute, Liege University; *Phng et al. (2013)*; PMID: 24046319 | ID_Zfin:ZDB-TGCONSTRCT-140610–8 | |
| Cell line (*Homo sapiens*) | HUVECs | Lonza | Lonza:C2519AS | Primary endothelial cells cultured as recommended by Lonza |
| Antibody | anti-CAMSAP2 (rabbit polyclonal) | Novus | Novus:NBP1-21402; RRID:AB_1659977 | (1:200) for IF; (1:1000) for WB |
| Antibody | anti-CEP135 (rabbit polyclonal) | Sigma-Aldrich | Sigma-Aldrich:SAB4503685; RRID:AB_10746232 | (1:300) |
| Antibody | anti-acetylated tubulin (rabbit polyclonal) | Sigma-Aldrich | Sigma-Aldrich:T7451; RRID:AB_609894 | (1:300) |
| Antibody | anti-polyglutamylated tubulin (rabbit polyclonal) | Sigma-Aldrich | Sigma-Aldrich:T9822; RRID:AB_477598 | (1:2000) |
| Antibody | anti- detyrosinated tubulin (rabbit polyclonal) | Abcam | Abcam:ab48389; RRID:AB_869990 | (1:2000) |
| Antibody | anti-γtubulin (rabbit polyclonal) | Sigma-Aldrich | Sigma-Aldrich:T3559, RRID:AB_477575 | (1:300) for IF; (1:1000) for WB |
| Antibody | anti-CDK5RAP2 (rabbit polyclonal) | Bethyl Laboratories | Bethyl Laboratories: A300-554A | (1:500) |
| Antibody | anti-EB3 (rabbit polyclonal) | *Stepanova et al., 2003*; PMID: 12684451 | | (1:400) |
| Antibody | anti-MMG8 (rabbit polyclonal) | *Wang et al. (2014)*; PMID: 25217626 | | (1:300) for IF; (1:1000) for WB |
| Antibody | anti-MYOSIN IIb (goat polyclonal) | Santa-Cruz biotechnology | Santa-Cruz biotechnology:sc-47205; RRID:AB_2297998 | (1:200) |
| Antibody | anti-PCM1 (goat polyclonal) | Santa-Cruz biotechnology | Santa-Cruz biotechnology:sc-50164; RRID:AB_2160195 | (1:300) |
| Antibody | anti-GM130 (mouse monoclonal) | BD Biosciences | BD Biosciences: 610823; RRID:AB_398142 | (1:600) |
| Antibody | anti-pericentrin (mouse monoclonal) | BD Biosciences | BD Biosciences: 611815; RRID:AB_399295 | (1:300) |
| Antibody | anti-EB1 (mouse monoclonal) | BD Biosciences | BD Biosciences: 610535; RRID:AB_397892 | (1:400) |
| Antibody | anti-VE-Cadherin (mouse monoclonal) | BD Biosciences | BD Biosciences: 610252; RRID:AB_2276073 | (1:500) |

*Continued on next page*

*Continued*

| Reagent type (species) or resource | Designation | Source or reference | Identifiers | Additional information |
|---|---|---|---|---|
| Antibody | anti-ZO-1 (mouse monoclonal) | BD Biosciences | BD Biosciences: 610966; RRID:AB_398279 | (1:200) |
| Antibody | anti-AKAP450 (mouse monoclonal) | BD Biosciences | BD Biosciences: 611518; RRID:AB_398978 | (1:300) for IF; (1:500) for WB |
| Antibody | anti-KU80 (mouse monoclonal) | BD Biosciences | BD Biosciences: 611360; RRID:AB_398882 | (1:3000) |
| Antibody | anti-CAMSAP3 (mouse monoclonal) | Sigma-Aldrich | Sigma-Aldrich: SAB4200415 | (1:500) |
| Antibody | anti-αtubulin (mouse monoclonal) | Sigma-Aldrich | Sigma-Aldrich:T5168; RRID:AB_477579 | (1:400) for IF; (1:2000) for WB |
| Antibody | anti-γtubulin (mouse monoclonal) | Sigma-Aldrich | Sigma-Aldrich: T6557; RRID:AB_477584 | (1:300) |
| Antibody | anti-NEDD1 (mouse monoclonal) | Abnova | Abnova:H00121441-M05; RRID:AB_534956 | (1:300) |
| Antibody | anti-αtubulin YL1/2 (rat monoclonal) | Pierce | Pierce: MA1-80017; RRID:AB_2210201 | (1:400) |
| Antibody | anti-CPAP (rabbit polyclonal) | *Kohlmaier et al. (2009)*; PMID: 19481460 | | (1:200) |
| Antibody | Alexa Fluor 488-, 594- and 647- secondaries | Molecular Probes | | (1:400) |
| Antibody | Alexa Fluor 488-,and 594- phalloidin | Molecular Probes | | (1:500) |
| Antibody | Abberior STAR 635P- anti-mouse | Sigma-Aldrich | Sigma-Aldrich:2-0002 -007-5 | (1:200) |
| Antibody | Atto 647N Phalloidin | Sigma-Aldrich | Sigma-Aldrich:65906 | (1:300) |
| Peptide, recombinant protein | VEGF-165 | Peprotech | Peprotech:100–20 | |
| Sequence-based reagent | siRNA against CAMSAP2#1 | *Jiang et al. (2014)*; PMID 24486153 | | 5'- GAATACTTCTTGACGAGTT-3' |
| Sequence-based reagent | siRNA against CAMSAP2#2 | *Jiang et al. (2014)*; PMID: 24486153 | | 5'- GTACTGGATAAATAAGGTA-3' |
| Sequence-based reagent | siRNA against CAMSAP3 | *Noordstra et al. (2016)*; PMID: 27802168 | | 5'-GCATTCTGGAGGAAATTGA-3' |
| Sequence-based reagent | siRNA against AKAP450 | *Hurtado et al. (2011)*; PMID: 21606206 | | 5'-AUAUGAACACAGCUUAUGA-3' |
| Sequence-based reagent | siRNA against MMG | *Roubin et al. (2013)*; PMID: 23430395 | | 5'-AGAGCGAGATCATGACTTA-3' |
| Sequence-based reagent | siRNA against CPAP | *Tang et al. (2009)*; PMID: 19503075 | | 5'- AGAAUUAGCUCGAAUAGAA-3' |
| Sequence-based reagent | morpholino against CAMSAP2b (Danio rerio) | Genetools | | ATACAGATGgcaagtcttttacatc |
| Sequence-based reagent | primers for CAMSAP2b (Danio rerio) amplification | This paper | | see sequences in the zebrafish experiment section |
| Commercial assay or kit | AMAXA huvecs nucleofector kit | Lonza | Lonza:VPB-1002 | |
| Recombinant DNA reagent | pLenti-RhoA2G | Addgene | Addgene:40179 | |
| Recombinant DNA reagent | pLVIN-Rac1-bs Rac1 | *Bouchet et al. (2016)*; PMID: 27939686 | | |
| Chemical compound, drug | rat tail collagen I | Corning | Corning:734–1085 | |
| Chemical compound, drug | Centrinone | *Wong et al. (2015)*; PMID: 25931445 | | |

*Continued on next page*

*Continued*

| Reagent type (species) or resource | Designation | Source or reference | Identifiers | Additional information |
|---|---|---|---|---|
| Chemical compound, drug | Y27632 | Sigma-Aldrich | Sigma-Aldrich:Y0503 | |
| Chemical compound, drug | Blebbistatin | Enzo Life Science | Enzo Life Science:BML-EI315-0005 | |
| Software, algorithm | ImageJ SOS plugin | *Yao et al., 2017*; PMID: 28324611 | | |
| Software, algorithm | ImageJ radiality plugin | https://github.com/ekatrukha/radialitymap | | *Katrukha, 2017*. radialitymap. Github. https://github.com/ekatrukha/radialitymap cf1e78f |
| Software, algorithm | imageJ curve tracing plugin | https://github.com/jalmar/CurveTracing | | *Teeuw and Katrukha, 2015*. CurveTracing. Github. https://github.com/jalmar/CurveTracing 960852 f |

## Antibodies, reagents and constructs

We used rabbit polyclonal antibodies against CAMSAP2 (Novus, Littleton, CO, NBP1-21402), CEP135, acetylated tubulin, polyglutamylated tubulin and γ-tubulin (Sigma-Aldrich, St Louis, MO, SAB4503685, T7451, T9822 and T3559), CDK5RAP2 (BethylLaboratories, Montgomery, TX, A300-554A), detyrosinated tubulin (Abcam, UK, ab48389), EB3 (*Stepanova et al., 2003*) and myomegalin isoform 8 (MMG8) (*Wang et al., 2014*), goat polyclonal antibodies against MYOSIN IIb and PCM1 (Santa-Cruz biotechnology, Dallas, TX, SC-47205 and SC-50164), mouse monoclonal antibodies against GM130, Pericentrin, EB1, VE-Cadherin, ZO-1, AKAP450 and KU80 (BD Biosciences, San Jose, CA, 610823, 611815, 610535, 610252, 610966, 611518 and 611360), CAMSAP3, α-tubulin and γ-tubulin (Sigma-Aldrich, SAB4200415, T5168 and T6557), NEDD1 (Abnova, Taiwan, H00121441-M05) and rat monoclonal antibodies against α-tubulin YL1/2 (Pierce, Waltham, MA, MA1-80017). Rabbit polyclonal antibody against CPAP was a kind gift of Dr. P.Gönczy (Swiss Institute for Experimental Cancer Research, EPFL, Lausanne, Switzerland).

For western blots, we used the following secondary antibodies: IRDye 800CW/680 LT Goat anti-rabbit and anti-mouse (Li-Cor Biosciences, Lincoln, LE). For immunofluorescence, Alexa Fluor 488,–594 and −647 conjugated goat antibodies against rabbit, rat and mouse IgG were used as secondary antibody (Molecular Probes, Eugene, OR), together with Alexa Fluor 488/594 phalloidin and NucRed Live 647 (Molecular Probes) and DAPI (Sigma-Aldrich).

For STED imaging, Atto 647N Phalloidin and Abberior STAR 635P anti-mouse antibodies (Sigma-Aldrich) were used.

High-concentration rat tail Collagen I was from Corning (Corning, NY), Phorbol 12-myristate 13-acetate (PMA), nocodazole, Y-27632 and DAPI were from Sigma-Aldrich and human recombinant Fibroblast Growth Factor (FGF) and Vascular Endothelial Growth Factor (VEGF) were from Peprotech (UK) and blebbistatin was from Enzo Life Sciences (Belgium). Centrinone was a kind gift of Dr. A. Shiau and Dr. T. Gahman (Small Molecule Discovery Program, Ludwig Institute for Cancer Research, San Diego).

The CAMSAP2 siRNA insensitive construct consists of a truncation of the first 232 amino acids of human CAMSAP2 generated by a PCR-based strategy and cloned into peGFP-C1 (Clonetech, Montain view, CA). The zebrafish rescue construct is described below in the zebrafish section.

The constructs coding for Rab6A and EB3 in peGFP-C2 were described elsewhere (*Matanis et al., 2002* and *Stepanova et al., 2003*), GFP-KIF13B was a gift from Dr.A.Chishti (University of Illinois, Chicago).

The Rho biosensor coding plasmid pLenti-RhoA2G (Addgene plasmid # 40179) is a gift of Dr.O. Pertz, (University of Basel, Switzerland) and the pLVIN-Rac1-bs Rac1 biosensor plasmid was described elsewhere (*Bouchet et al., 2016*).

## siRNAs

We used the following siRNAs purchased from Sigma siRNA CAMSAP2 #1, 5'- GAATACTTCTTGAC-GAGTT-3' (*Jiang et al., 2014*) siRNA CAMSAP2 #2, 5'- GTACTGGATAAATAAGGTA-3' (*Jiang et al., 2014*) siRNA CAMSAP3, 5'-GCATTCTGGAGGAAATTGA-3' (*Noordstra et al., 2016*) siRNA AKAP450, 5'-AUAUGAACACAGCUUAUGA-3' (*Hurtado et al., 2011*) siRNA MMG, 5'-AGAGCGAGATCATGACTTA-3' (*Roubin et al., 2013*) siRNA CPAP, 5'- AGAAUUAGCUCGAA UAGAA-3' (*Tang et al., 2009*) siRNA Luciferase control, 5'-CGTACGCGGAATACTTCGA-3'

## Cell culture and treatment

Human Umbilical Vein Endothelial Cells (HUVECs) were obtained from Lonza and grown in endothelial basal medium (EGM-2) supplemented with growth supplements (SingleQuots, Lonza, Switzerland): 2% Fetal Bovine Serum (FBS), human Epidermal Growth Factor (hEGF), Vascular Endothelial Growth Factor (VEGF), R3-Insulin-like Growth Factor-1 (R3-IGF-1), Ascorbic Acid, Hydrocortisone human Fibroblast Growth Factor-Beta (hFGF-β), Heparin, Gentamicin/Amphotericin-B (GA).

HUVECs authentication was guaranteed by Lonza through identity and quality control and testing, including against mycoplasma, bacteria, yeast, and fungi. Only low passage cells (between passages 3 and 7) were used.

Plasmids and siRNA were, respectively, nucleofected using Amaxa technologies with the HUVEC nucleofector kit (Lonza) and transfected with GeneTrans II (MoBiTec, Germany) reagents according to the manufacturers' protocols. CPAP depletion was achieved through four successive rounds of siRNA transfection every 3 days.

Amaxa was additionally used in the specific case of siRNA transfection of centrinone-treated HUVECs.

HUVECs were treated with 125 nM centrinone for 9 days, including the time needed for functional assay. During treatment, non-treated and centrinone-treated HUVECs were passaged every 2–3 days to keep a 60–90% confluency. All functional assays involving centrinone were done in the presence of 10 nM Thymidine.

For VEGF treatment, HUVECs were starved in 0.5% serum containing medium for 36 hr before addition of 50 ng/ml of VEGF.

Nocodazole-induced MT complete disassembly was performed by treating HUVECs with 10 µM nocodazole for 2 hr at 37°C, followed by 1 hr at 4°C. Washout (WO) was then carried out by two washes with cold and two washes with warm medium.

Y-27632 and blebbistatin were used at 10 µM and 50 µM, respectively.

## Cell proliferation assays

HUVECs were trypsinized 48 hr after transfection and counted by trypan blue staining for quantification of the doubling time. Alternatively, HUVECs were stained with phalloidin together with DAPI to identify mitotic cells and their percentage was calculated to determine the mitotic index.

## Western blotting

HUVECs total extracts were prepared in RIPA buffer (10 mM Tris-HCl pH 8, 140 mM NaCl, 1 mM EDTA, 1 mM EGTA, 1% Triton X-100, 0,1% SDS, protease inhibitor cocktail (Complete - Sigma)).

SDS-PAGE and Western blot analysis were performed according to standard procedures and developed with the Odyssey technology (Li-Cor Biosciences). Densitometric analysis was done using the 'gel analysis' plug-in of ImageJ.

## Monolayer wound healing assay

A confluent HUVEC monolayer was scratched using a sterile P200 tip to create a cell-free zone. Fields were photographed just after injury and 8 hr later. Quantification of cell migration was made by measuring the percentage of area recovery using ImageJ software. Alternatively, phase-contrast live imaging was performed.

## 3D individual seeding assay

Single HUVECs were seeded into 2.5 mg/ml collagen pH buffered gels overlaid with complete medium supplemented with 50 ng/ml FGF, VEGF and PMA. For global vascular network assessment,

$2 \times 10^6$ cells/ml were embedded for 48 hr and wide-field fluorescence imaging was done on EVOS cell imaging system (ThermoFisher Scientific, Waltham, MA). For cell morphology analyses, $0.5 \times 10^6$ cells/ml embedding experiments were submitted to phase-contrast live imaging or processed for immunostaining after 24 hours.

## 3D spheroid sprouting assay

The spheroid sprouting assay was performed as previously described (*Martin et al., 2013*): HUVEC spheroids were generated overnight by culturing endothelial cells in complete medium containing 20% methylcellulose in non-adherent 96 well plates. Harvested spheroids were then embedded into 2 mg/ml collagen pH buffered gels overlaid with complete medium supplemented with 40 ng/ml FGF and 50 ng/ml PMA. Angiogenic activity was quantified by measuring the cumulative length of the sprouts that had grown out of each spheroid, their mean number and length, 24 hr after embedding using ImageJ software on bright field images. The sprouts that were originating from secondary branching and the ones that were not in focus in the pictures were omitted from the analysis.

## Immunofluorescence staining

For 2D (/3D) staining, HUVECs were fixed with −20℃ methanol for 10 min or with 4% PFA for 12 (/20) minutes at RT, permeabilized with 0.15% Triton X-100 in phosphate buffered saline (PBS) (/PBS-glycine 0.1M) for 2 (/45) minutes, sequentially incubated 1 hr in blocking buffer 2% BSA, 0.05% Tween-20 in PBS (/2% BSA, 0.05% Tween-20, 0.2% Triton X-100, 0.05% $NaN_3$ in PBS), 1 (/4) hr in primary antibody in blocking buffer, 1 (/1.5) hour in secondary antibody, Alexa-conjugated phalloidin and DAPI in blocking buffer. After several washes, slides (/dismounted 3D gel plugs) were air-dried and mounted in Vectashield mounting medium (Vector laboratories, Burlingame, CA).

For immunostaining of HUVECs submitted to wound-healing assay, samples were fixed after 6 hr of migration.

For STED imaging, HUVECs were pre-extracted 45 s in extraction buffer (PEM80, 0.3% Triton-X100, 0.15% gluteraldehyde) at 37℃, fixed with 4% PFA for 12 min at 37℃, permeabilized with 0.2% Triton X-100 for 10 min at RT and then submitted to the same 2D protocol as above except the removal of Tween-20 in the buffers.

## Image acquisition

Bright-field images were collected on an EVOS cell imaging system (ThermoFisher Scientific) and phase-contrast live cell imaging was performed on a Nikon Ti equipped with a perfect focus system Nikon), a super high pressure mercury lamp (C-SHG1, Nikon, Japan), a Plan Fluor DLL 10x NA 0.3 (Ph1), a CoolSNAP HQ2 CCD camera (Photometrics, Tucson, AZ), a motorized stage MS-2000-XYZ with Piezo Top Plate (ASI, Eugene, OR) and a stage top incubator (Tokai Hit, Japan) for 37℃/5% $CO_2$ incubation. The microscope setup was controlled by Micro-manager software.

For fluorescence imaging of 2D fixed samples and 3D fixed and live samples, including zebrafishes, Z-series images were collected with spinning disk confocal microscopy on a Nikon Eclipse Ti microscope equipped with a perfect focus system (Nikon), a spinning disk-based confocal scanner unit (CSU-X1-A1, Yokogawa, Japan), an Evolve 512 EMCCD camera (Roper Scientific, Trenton, NJ) attached to a 2.0X intermediate lens (Edmund Optics, Barrington, NJ), a super high pressure mercury lamp (C-SHG1, Nikon), a Roper scientific custom-ordered illuminator (Nikon, MEY10021) including 405 nm (100 mW, Vortran), 491 nm (100 mW, Cobolt) 561 nm (100 mW, Cobolt) and 647 nm (100 mW, Cobolt) excitation lasers, a set of BFP, GFP, RFP and FarRed emission filters (Chroma, Bellows Falls, VT) and a motorized stage MS-2000-XYZ with Piezo Top Plate (ASI). The microscope setup was controlled by MetaMorph. Images were acquired using Plan Fluor 20x MI NA 0.75 and Plan Apo VC 60x NA 1.4 oil objectives and Apo LWD λS 40x water immersion objective. When necessary, a stage top incubator maintaining 37℃ or 28℃ and 5% $CO_2$ was used.

Live Fluorescence imaging of EB3-GFP in 2D and GFP-Rab6A in 3D was performed on the same spinning disk confocal configuration. Acquisitions were performed at five frames/s during 2 min.

Alternatively, 2D samples imaging was performed using widefield fluorescence illumination on a Nikon Eclipse 80i upright microscope equipped with a CoolSNAP HQ2 CCD camera (Photometrics), an Intensilight C-HGFI precentered fiber illuminator (Nikon), a Plan Apo VC 100x NA 1.4 oil or 60x NA 1.4 oil and driven by Nikon NIS Br software.

Live-cell TIRF imaging was performed on a Nikon Eclipse Ti-E inverted microscope equipped with perfect focus system (Nikon), a CFI Apo TIRF 100X oil objective (Nikon), a TI-TIRF-E motorized TIRF illuminator (Nikon), a QuantEM 512SC EMCCD camera (Photometrics, Roper Scientific) and a stage top incubator maintaining 37°C and 5% $CO_2$ (Tokai hit). The system was controlled with MetaMorph 7.5 software (Molecular Devices, San Jose, CA).

Gated STED imaging was performed with Leica TCS SP8 STED 3X microscope driven by LAS X controlling software and using HC PL APO 100x/1.4 oil STED WHITE objective, 633 nm white laser for excitation and 775 nm pulsed lased for depletion. Images were acquired in 2D STED mode with vortex phase mask. Depletion laser power was equal to 90% of maximum power and an internal Leica GaAsP HyD hybrid detector with a time gate of $1 \leq tg \leq 8$ ns was used.

## FRET analysis of RhoA and Rac1 biosensors

The activities of RhoA and Rac1 were measured using previously described Rho single chain biosensor (*Fritz et al., 2013*) and Rac1 single chain biosensor (*Moshfegh et al., 2014*) using ratiometric FRET between mTFP1/mCerulean and mVenus. Live FRET imaging was performed on Leica TCS SP8 microscope equipped with spectral detection using HC PL APO 100x/1.4 oil STED WHITE objective. 440 nm pulsed laser (40MHz) was used for the excitation. Two channels were acquired simultaneously using hybrid detectors in the spectral ranges of donor 450–500 nm and acceptor 515–550 nm. Images were acquired with a scanning velocity of 100 Hz and eight line average scans, a pixel size of 0,416 μm and dimensions of 256 × 256 pixels. Donor and acceptor images were convolved with a Gaussian of 1 pixel, background subtracted and FRET-index image was calculated using ImageJ macro according to the formula: $FRET = \frac{I_A}{I_A + I_D}$,

where $I_A$ and $I_D$ correspond to the pixel intensity values of acceptor and donor images. Cell outlines were determined from the thresholded acceptor image and the average FRET-index value was calculated per cell.

## Image preparation and analysis

For image preparation, we used ImageJ for adjustments of levels and contrast, maximum intensity projections, stitching (with pairwise stitching plugin) and thresholding to create binary mask used for circularity measurements and particle detection.

Kymographs of MT plus end and Rab6 3D dynamics were made using the KymoResliceWide plugin of ImageJ software and analyzed using the same software. For MTs, only length changes $\geq 0.3$ μm between two consecutive time points were considered as growth or shortening events, while changes $< 0.3$ μm were considered as a pause event; only the events starting and finishing within the recording were analyzed. Velocity was calculated for each growth event and then averaged. Catastrophe frequency was calculated by dividing the number of catastrophes (transition from growth or pause to shortening) by the sum of growth and pause durations.

EB3 comets and CAMSAP2 stretches were automatically detected on thresholded pictures using the Particle Analysis plugin of ImageJ software. Their number (EB3) and surface (CAMSAP2) were quantified and reported to the cell surface area.

EB3 enrichment at the Golgi (/centrosome) was calculated as the ratio between the average EB3 intensity in a 2 μm diameter-circle drawn around Golgi mini-stacks (/centrosome) and the average EB3 intensity in the cytoplasm.

Golgi dispersion was calculated as $\frac{SDi}{\sqrt{i}}$ in which $SD_i$ is the standard deviation of intensity and i the mean intensity.

For cell-cell junctions analysis (VE-Cadherin and ZO-1), ImageJ was used to plot intensity profiles along a manually drawn line across junctions. These profiles were then analyzed using the 'area under curve' function of GraphPad prism five and the maximum value of, as well as the area under the peaks were averaged.

F-actin staining in 3D was analyzed similarly by plotting intensity profiles along a 10 μm long rectangle drawn 5 μm away from the cell body using ImageJ and measuring the maximum intensity and the peak area using GraphPad prism 5. For stress fiber analysis, a customized ImageJ macro was used to trace the stress fibers and measure their length and width (*Teeuw and Katrukha, 2015*) (available at https://github.com/jalmar/Curve Tracing).

All cell protrusions were manually traced with ImageJ software to quantify their number, length and spatial distribution. Polarity index was calculated as $\frac{\sum(1-|\sin\alpha_i|).\ L_i}{\sum L_i}$ in which $\alpha_i$ is the angle between the protrusion$_i$ and the longest protrusion and $L_i$ is the length of the protrusion. Cell masks were analyzed using the Particle Analysis plugin of ImageJ software to measure cell circularity, calculated as $\frac{4\pi A}{P^2}$ in which A and P are the area and the perimeter of the cell mask, respectively.

Time-lapse imaging of monolayer wound healing assays was analyzed using Manual Tracking and Chemotaxis Tool plugin of ImageJ software to measure the velocity and directionality (the ratio between the Euclidian and accumulated distance) of cell movement.

Time-lapse imaging of Rab6 and KIF13B was analyzed using ImageJ software with a recently developed plugin (*Yao et al., 2017*). The SOS plugin combined two procedures: particle detection and particle linking: SOS detector 3D module as detector and SOS linker (NGMA) module as linker were used. The tracking results were then processed using MTJ (MTrackJ) Simple Track Segment module to remove the non-directional tracks and to analyse speed, duration and length of the runs. MTJ Measure Region was used to determine the number of tracks contained in the cell front area, defined as the 45° sector facing the cell lamellipodia and originating from the center of the nucleus. Percentage of tracks in the front was calculated by dividing this value by the total number of tracks and correcting for area differences.

Radial representations of time-lapse images of protrusion formation in 3D were made using successively the Radial Reslice, Reslice and Minimum Intensity projection functions of ImageJ software.

ImageJ Radial Profile plugin was used to measure the distribution of CAMSAP2 and GM130 signal intensity along the radius in a 20 μm radius-circle originating from the Golgi center and each profile was normalized as $\frac{x\ i-MAXxi}{MAXxi-MINxi}$.

For Golgi enrichment index of α- or γ-tubulin, z-maximum projection of α-, γ-tubulin or GM130 channel was thresholded using ImageJ to create a binary mask to delineate the cell or the Golgi area. The difference between the average intensity within the Golgi area and in the area outside the Golgi was divided by the intensity outside the Golgi and expressed in percent.

α-tubulin or CAMSAP2 enrichment in the longest protrusion was calculated as the ratio between the fluorescence intensity in a 0.75 μm diameter-circle drawn 8 μm away from the cell body or in the manually drawn area in the longest protrusion and in the other protrusions, within z-maximum projections and averaged per cell.

Time-lapse imaging of zebrafish venous sprouting was analyzed by manually drawing the vector corresponding to a sprout for each time point and measuring sprout length and angle using the measure function of ImageJ software. When needed, 3D color-coded stacks were used to more easily isolate the venous sprout. When the geometry of a sprout did not fit a straight line, a segmented line was used for length measurement and the straight line between sprout extremities for the angle determination. Growth persistence of a sprout elongation event corresponds to the mean value of length variations (Δ Length) between two consecutive time frames (every 10 min), whereas directional persistence was calculated for each frame as $\frac{1}{\sin|\alpha|}$, where Δα represents the angle variation between two consecutive time frames, and then averaged per growing event.

To analyze MT radiality, images of fluorescently labeled MTs were separated into radial and non-radial components using customized ImageJ macro (*Katrukha, 2017*). (available at https://github.com/ekatrukha/radialitymap; copy archived at https://github.com/elifesciences-publications/radialitymap). First, local orientation angle map was calculated for each pixel using OrientationJ plugin (*Püspöki et al., 2016*). We used 'cubic spline gradient' method and tensor sigma parameter of 6 pixels (0.4 μm). The new origin of coordinates was specified by selecting the centrosome position in a corresponding channel, or the brightest spot in case of centrinone treatment. Radial local orientation angle was calculated as a difference between the local orientation angle and the angle of vector drawn from the new origin of coordinates to the current pixel position. A radial map image was calculated then as an absolute value of the cosine of the radial local orientation angle at each pixel providing values between zero and one. A non-radial map image was calculated as one minus radial map. Both maps were multiplied with the original image to account for different signal intensities; the two maps illustrate separated radial and non-radial image components.

The radial profile of the signal in the non-radial map image (normalized to the maximum signal of the original picture) was built using ImageJ and used to calculate the average non-radial proportion

of the MT network. To avoid the artifacts of the cell center (very high signal) and border (MT bending), only a circular section around the reference point was used in the averaging (from 2.5 µm to 15 µm).

All mentioned ImageJ plugins have source code available and are licensed under open-source GNU GPL v3 license.

## Zebrafish experiments

The *Tg(fli1a:eGFP)y1* and *Tg(Fli1ep:Lifeact-EGFP)* (*Phng et al., 2013*) lines were raised according to EU regulations on laboratory animals. All animal experiments were approved by the animal welfare committee of the University of Liege (protocol number 14–1556, laboratory agreement number LA 1610002). Knockdown experiments were performed by injecting embryos at the one- to two-cell stage with 6 ng of Camsap2b morpholino. The following Camsap2b-splice blocking morpholino sequence was used: ATACAGATGgcaagtcttttacatc.

For rescue experiment, a Morpholino insensitive human Camsap2 was built by overlapping PCR-based strategy (ATACAGATG transformed into ATTCAAATG), inserted into PSC2 + vector, linearized, in vitro transcribed and injected at 50 ng/µl.

For RT-PCR, cDNA was generated from total RNA extracted from zebrafish embryos with Trizol reagent (Thermo Fisher Scientific) using RevertAid RT Kit (Thermo Fisher Scientific) with random hexamer primers. After DNAse treatment (Thermo Fisher Scientific), cDNA was submitted PCR amplification followed by gel electrophoresis analysis using the following primers: AC TTCAGCAGGGCCAAGATA and TGTCACAGCCTCTTCAGCAT.

Alternatively, cDNA was submitted to quantitative real-time (q)PCR using Sybrgreen technology (Applied Biosystems, Foster City, CA) on a ViiA7 apparatus (Applied Biosystems). ELFA was used as reference gene to quantify the relative expression of the exon2 of Camsap2b using the $\Delta\Delta$Ct method with three alternative primers pairs. Primer sequences were as followed

F1_CAMSAP2b, GTCATAACGCCGTCATCCAG
R1/2_CAMSAP2b, TGTATAGGGGTCTTGCAGAGG
F2_CAMSAP2b, GGGGAGTCTGATTCTCAGGA
F3_CAMSAP2b, ACTTCAGCAGGGCCAAGATA
R3_CAMSAP2b, TCACGAGTCTCTCCTGGTCA
F_ELFA, CTTCTCAGGCTGACTGTGC
R_ELFA, CCGCTAGCATTACCCTCC

For analyses of vascular structure formation, screening was performed under a fluorescence stereomicroscope whereas confocal pictures and movies were performed on artificially dechorionated embryos between 30 and 48hpf embedded in low melting point agarose (0.8%).

## Statistics

Statistical analyses were performed using GraphPad Prism five or Excel and significance was assessed using Mann-Whitney U-, Chi square with Yates correction- and Student's t- two-tailed paired and unpaired tests. The statistical test used as well as the sample size is indicated in the figure legends. All data are shown using box plots where rectangles represent the second and third quartiles, contain a line corresponding to the median value and are extended with whiskers showing the minimum and maximum, except in *Figure 6—figure supplement 1B* and *Figure 9D*, which depict mean ± SEM. In *Figure 1—figure supplement 1B* and *Figure 4—figure supplement 1B*, the mean value ± SEM is indicated within the pictures. No explicit power analysis was used to determine sample size and no masking was used for analysis.

## Additional information

### Competing interests

Anna Akhmanova: Senior editor, *eLife*. The other authors declare that no competing interests exist.

## Funding

| Funder | Grant reference number | Author |
| --- | --- | --- |
| European Research Council | Synergy 609822 | Anna Akhmanova |
| Nederlandse Organisatie voor Wetenschappelijk Onderzoek | ALW Open Program grant 824.15.017 | Anna Akhmanova |
| H2020 Marie Skłodowska-Curie Actions | IEF fellowship | Maud Martin |
| China Scholarship Council | PhD fellowship | Jingchao Wu |
| Fonds De La Recherche Scientifique - FNRS | FRIA fellowship | Alexandra Veloso |

The funders had no role in study design, data collection and interpretation, or the decision to submit the work for publication.

## Author contributions

Maud Martin, Conceptualization, Data curation, Formal analysis, Funding acquisition, Investigation, Visualization, Methodology, Writing—original draft, Project administration, Writing—review and editing; Alexandra Veloso, Formal analysis, Funding acquisition, Investigation, Visualization, Writing—original draft; Jingchao Wu, Conceptualization, Resources, Funding acquisition, Methodology; Eugene A Katrukha, Software, Formal analysis, Methodology; Anna Akhmanova, Conceptualization, Supervision, Funding acquisition, Writing—original draft, Project administration, Writing—review and editing

## Author ORCIDs

Maud Martin http://orcid.org/0000-0003-0048-6437
Anna Akhmanova http://orcid.org/0000-0002-9048-8614

## Ethics

Animal experimentation: All animal experiments were approved by the animal welfare committee of the University of Liege (protocol number 14-1556, laboratory agreement number LA 1610002).

## Decision letter and Author response

Decision letter https://doi.org/10.7554/eLife.33864.052
Author response https://doi.org/10.7554/eLife.33864.053

## Additional files

### Supplementary files

• Transparent reporting form
DOI: https://doi.org/10.7554/eLife.33864.050

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
