## [Decision Letter]

Thank you for submitting your article "Control of endothelial cell polarity and sprouting angiogenesis by non-centrosomal microtubules" for consideration by *eLife*. Your article has been reviewed by three peer reviewers, and the evaluation has been overseen by Fiona Watt as the Senior Editor and Pekka Lappalainen as the Reviewing Editor. The reviewers have opted to remain anonymous.

The reviewers have discussed the reviews with one another and the Reviewing Editor has drafted this decision to help you prepare a revised submission.

Summary:

This manuscript focuses on the roles of different types of microtubule anchoring/stabilization mechanisms (centrosomally anchored versus Golgi-anchored CAMSAP2-stabilized versus CAMPSAP2-stabilized not Golgi-anchored) in the polarization and movement of endothelial cells in 2D and 3D environments. The major finding is that, especially in soft 3D environments, centrosomal microtubules are not essential for establishment of polarity, whereas CAMSAP2-stabilized, non-centrosomal microtubules are required for persistent elongation of the cell body. Moreover, the authors show that CAMSAP2 plays a role in sprouting angiogenesis in vivo.

These findings are interesting, and the analysis and quantification methods are well designed and state of the art. The topic of the interplay between centrosomal, Golgi-anchored, and non-centrosomal microtubules and the role of the CAMSAPs in the dynamics of non-centrosomal arrays is also of high current interest. However, there are few points that should be addressed to further strengthen the study.

Essential revisions:

1) The authors should examine at which level the cell morphology is affected in the absence of CAMSAPs (i.e. do CAMSAP-dependent microtubules generate and stabilize protrusions in ECs or do they stabilize protrusions indirectly by regulating the actin cytoskeleton). The authors should thus more thoroughly examine the organization and dynamics of the actin cytoskeleton in control vs. CAMSAP knockdown cells. Because CAMSAP knockdown cells appear to have more stress fibers compared to control cells (based on the images presented in Figures 4B and C), the authors could also use e.g. blebbistatin to examine whether the differences in the number and persistence of protrusions between control and CAMSAP knockdown cells could result from their different contractility.

2) Unlike stated in the third paragraph of the Discussion, the authors cannot conclude from their data that CAMSAP2 depletion completely blocks sprouting from spheroids. There are plenty of short sprouts (in line with protrusive activity of isolated ECs, which can make protrusions), but not very long, single ones.

---

## [Author Response]

Essential revisions:1) The authors should examine at which level the cell morphology is affected in the absence of CAMSAPs (i.e. do CAMSAP-dependent microtubules generate and stabilize protrusions in ECs or do they stabilize protrusions indirectly by regulating the actin cytoskeleton). The authors should thus more thoroughly examine the organization and dynamics of the actin cytoskeleton in control vs. CAMSAP knockdown cells. Because CAMSAP knockdown cells appear to have more stress fibers compared to control cells (based on the images presented in Figures 4B and C), the authors could also use e.g. blebbistatin to examine whether the differences in the number and persistence of protrusions between control and CAMSAP knockdown cells could result from their different contractility.

It is true that there is a tight interplay between the microtubule and the acto-myosin cytoskeleton that regulates cell shape and motility. In particular, the effect of microtubule signaling on Rho GTPase-mediated control of cell protrusion and contractility is often thought to represent the most important microtubule function during cell migration.

In order to find out whether the role of CAMSAP2-stabilised non-centrosomal microtubules was mediated by the regulation of the acto-myosin cytoskeleton, we performed a new set of 2D and 3D experiments to:

- assess the impact of CAMSAP2 depletion on the actin cytoskeleton

- examine whether a potential change in contractility could explain CAMSAP2-associated defects.

The entire new Figure 4 (complemented with two supplement figures) now addresses this question in detail. We found that whereas CAMSAP2-knockdown cells exhibited a small increase in the abundance of stress fibers in 2D (Figure 4A, Figure 4—figure supplement 1A), this change was not associated with alterations in Rho and Rac1 activation pattern, the arrangement of cell-cell junctions or formation of myosin IIb-enriched retracting rear (Figure 4B, Figure 4—figure supplement 1B, C, already included the first version of the manuscript). More importantly, when looking in the context of a 3D environment, we now found that within the protrusions, the actin cytoskeleton that is mainly cortical was unaltered following CAMSAP2 depletion (Figure 4C, Figure 4—figure supplement 1D).

In agreement with previous 3D observations showing that cortical contractility counteracts protrusion formation (Sanz-Moreno et al., 2008, Fischer et al., 2009, Bouchet et al., 2016), we observed longer and more persistent protrusions, as well as more and longer spheroid sprouts after treatment with the ROCK inhibitor Y-27632 and the myosin II inhibitor blebbistatin (Figure 4E, F, Figure 4—figure supplement 2A-F). Nevertheless, even in a background with reduced contractility, CAMSAP2 silencing still severely affected protrusion persistence, cell elongation and sprout formation (Figure 4D, E, F, Figure 4—figure supplement 2A-F). Taken together, these data show that CAMSAP2-stabilised microtubules are controlling protrusion persistence and organization without significantly affecting acto-myosin cytoskeleton and contractility, and we added this consideration to the Discussion (second paragraph).

2) Unlike stated in the third paragraph of the Discussion, the authors cannot conclude from their data that CAMSAP2 depletion completely blocks sprouting from spheroids. There are plenty of short sprouts (in line with protrusive activity of isolated ECs, which can make protrusions), but not very long, single ones.

We agree that CAMSAP2-depleted spheroids could still form short sprouts and that saying that CAMSAP2 silencing almost completely blocked sprouting is a too strong statement. In total agreement with the reviewers’ comment, we think that the “protrusive activity” of CAMSAP2depleted ECs, important for protrusion initiation, is not affected, explaining the presence of short sprouts. Nevertheless, whereas the lack of protrusion persistence in isolated ECs “only” impaired their organization, the inability of CAMSAP2-silenced spheroids to mature larger sprouts caused their regression (Figure 3—figure supplement 1A). This ultimately led to a difference in the severity of the phenotypes that we do find striking (for example, if we compare Figure 2G with Figure 3—figure supplement 1C).

We rephrased our statement to more accurately reflect these considerations (subsection “CAMSAP2 is required for maintaining non-centrosomal MTs and cell migration in ECs”, last paragraph; subsection “CAMSAP2 is required for stabilization of one major cell protrusion”, first paragraph; subsection “Centrosome removal promotes cell polarization in the absence of CAMSAP2”, last paragraph; Discussion, second and third paragraphs).